# Pressure and inertia sensing drifters for glacial hydrology flow path measurements

Andreas Alexander[1, 2], Maarja Kruusmaa[3, 4], Jeffrey A. Tuhtan[3], Andrew J. Hodson[2, 5], Thomas V. Schuler[1, 6], and Andreas Kääb[1]

[1]Department of Geosciences, University of Oslo, 0316 Oslo, Norway
[2]Department of Arctic Geology, The University Centre in Svalbard, 9171 Longyearbyen, Norway
[3]Centre for Biorobotics, Tallinn University of Technology, 12618 Tallinn, Estonia
[4]Centre for Autonomous Marine Operations and Systems, Norwegian University of Science and Technology, 7491 Trondheim, Norway
[5]Department of Environmental Sciences, Western Norway University of Applied Sciences, 6856 Sogndal, Norway
[6]Department of Arctic Geophysics, The University Centre in Svalbard, 9171 Longyearbyen, Norway

**Correspondence:** Andreas Alexander (andreas.alexander@geo.uio.no)

**Abstract.** Glacial hydrology plays an important role in the control of glacier dynamics, sediment transport and of fjord and proglacial ecosystems. Surface meltwater drains through glaciers via supraglacial, englacial and subglacial systems. Due to challenging field conditions, the processes driving surface processes in glacial hydrology remain sparsely studied. Recently, sensing drifters have shown promise in river, coastal and oceanographic studies. However, practical experience with drifters in glacial hydrology remains limited. Before drifters can be used as general tools in glacial studies, it is necessary to quantify the variability of their measurements. To address this, we conducted repeated field experiments in a 450 m long supraglacial channel with small cylindrical drifters equipped with pressure, magnetometer, acceleration and rotation rate sensors and compared the results. The experiments (n = 55) in the supraglacial channel show that the pressure sensors consistently yielded the most accurate data, where values remained within ±0.11 % of the total pressure time-averaged mean (95% confidence interval). Magnetometer readings also exhibited low variability across deployments, maintaining readings within ±2.45% of the time-averaged mean of the magnetometer magnitudes. Linear acceleration measurements were found to have a substantially higher variability of ±34.4% of the time-averaged mean magnitude and the calculated speeds remained within ±24.5% of the time-averaged mean along the flow path. Furthermore, our results indicate that prominent shapes in the sensor records are likely to be linked to variations in channel morphology and associated flow field. Our results show that multi-modal drifters can be a useful tool for field measurements inside supraglacial channels. Future deployments of drifters into englacial and subglacial channels promise new opportunities for determining hydraulic and morphologic conditions from repeated measurements of such inaccessible environments.

## 1 Introduction

Glacial hydrology plays a key role in glacier dynamics (Flowers, 2018), sediment transport and its impact on fjord and proglacial ecosystems (e.g. Swift et al., 2005; Meire et al., 2017; Urbanski et al., 2017). Surface water is generally routed

supraglacially, i.e. along the glacier surface in ice-walled drainage systems. Water enters the englacial and subglacial drainage system through moulins, crevasses and cut-and-closure systems (Gulley et al., 2009). Ice-walled drainage systems have highly variable geometry, controlled by the counteracting mechanisms of melt enlargement due to dissipation of potential energy and creep-closure of the viscous ice (Röthlisberger, 1972). The capacity of the glacial drainage system varies in both space

and time and dynamically adjusts to the highly variable meltwater supply (Schoof, 2010; Bartholomew et al., 2012). These geometric adjustments often form step-pool sequences (e.g. Vatne and Irvine-Fynn, 2016) and are responsible for up to 90% of the total flow resistance (Curran and Wohl, 2003). The channel geometry influences flow resistance and water velocity (Germain and Moorman, 2016). Vice versa, the velocity controls the incision rates in ice-walled channels in conjunction with water temperature and the rate of heat loss at channel boundaries (Lock, 1990; Isenko et al., 2005; Jarosch and Gudmundsson,

2012). Despite these findings, major knowledge gaps remain, especially within subglacial hydrology due to limited observations of the environment. Specifically, the mechanisms driving water routing from the glacier surface to the bed remain largely unexplored. Improving our understanding of glacial hydrology and its effect on glacier dynamics requires new methods. The methods should be able to provide direct measurements of water routing on, through and under glaciers including the water temperature, velocity, pressure as well as the channel morphology along multiple flow paths.

Direct measurements are the most ideal source of information, but remain scarce in glacial hydrology because they are difficult to obtain (e.g. Gleason et al., 2016). Beginning in the early 2000s, new technologies have emerged, and current methods for in-situ tests include among others Doppler current profiling in supraglacial systems (e.g. Gleason et al., 2016), dye tracing (e.g. Seaberg et al., 1988; Willis et al., 1990; Fountain, 1993; Nienow et al., 1998; Hasnain et al., 2001; Schuler and Fischer,

2009), salt injection gauging (e.g. Willis et al., 2012), geophysical methods (e.g. Diez et al., 2019) and gas tracing (e.g. Chandler et al., 2013).

Lagrangian drifters are small floating devices, which passively follow the water flow and are commonly used to study flow in large rivers, lakes and oceans. Most typically, drifters provide information about their position and speed (Landon et al.,

2014). Depending on sensor payload, drifters can be used for a wide range of applications, including coastal and ocean surface current monitoring (Boydstun et al., 2015; Jaffe et al., 2017), to estimate river bathymetry and surface velocities (Landon et al., 2014; Almeida et al., 2017) and to collect imagery for underwater photogrammetry (Boydstun et al., 2015). Recent payloads in river studies included sensors for temperature (e.g. Tinka et al., 2013; Oroza et al., 2013; Allegretti, 2014), dissolved oxygen (e.g. D'Este et al., 2012; Tinka et al., 2013), pH (e.g. Tinka et al., 2013; Arai et al., 2014), turbidity (e.g. Marchant et al.,

2015), as well as GPS receivers (e.g. Stockdale et al., 2008; Tinka et al., 2013), Acoustic Doppler Current profilers (e.g. Tinka et al., 2009; Postacchini et al., 2016) and Inertial Measurement Units (e.g. Arai et al., 2014). Additionally, sensor payloads in oceanographic applications included devices for the measurement of conductivity (e.g. Reverdin et al., 2010; Jaffe et al., 2017), chlorophyll (e.g. Jaffe et al., 2017) and underwater imagery (e.g. Boydstun et al., 2015; Xanthidis et al., 2016). Drifters remain the most promising method for the study of physical parameters along multiple flow paths within the hydrological system of

a glacier. This is because Lagrangian drifters can be equipped with multiple sensors to collect data along the flow path within

the changing environment. Therefore they provide a wider range of observational data with reduced deployment effort when compared to conventional fixed station hydrological measurements.

Development of sensing drifters in glaciology has been previously reported, most notably the Moulin Explorer by Behar et al. (2009) which was unfortunately lost during its first deployment. A successful glacial drifter was the E-tracer, as reported by Bagshaw et al. (2012). The device has the size of a table tennis ball and included a radio transmitter to enable identification and data transmission after passage through the subglacial system of Leverett Glacier, Greenland (Bagshaw et al., 2012). These encouraging results lead to a second generation of E-tracers equipped with a pressure sensor, and successfully transmitted pressure data from subglacial channels through 100 meters of overlying ice after having been deployed in crevasses and moulins (Bagshaw et al., 2014). The published data remains however sparse and limited to a single mean pressure record in Bagshaw et al. (2014). As with all new field measurement technologies, the repeatability of in-situ measurements is often very challenging to determine. Encouraged by the previous drifter studies by Bagshaw et al. (2012) and Bagshaw et al. (2014) with a single pressure sensor, the present study explores the potential of sensing drifters with several different sensors (multi-modal) to record data along the flow path of glacier channels. The focus of this work is to assess the repeatability of Lagrangian drifter measurements in a supraglacial channel. Current methods, such as dye tracing, allow for the repeatable measurement of the flow velocities averaged over the duration of a passage. However, it is impossible to deconvolve these records to obtain spatially and temporally distributed information. The present study therefore also assesses the potential of sensing drifters to acquire spatial and temporal variation of the velocity along a flow path. Furthermore, the multi-modal sensor data are investigated for potential time series features that may be associated with geometrical features of the investigated supraglacial channel.

The experiments were carried out using a submersible multi-modal drifter platform measuring at 100 Hz. The rugged, autonomous sensing platform records the water pressure, and three components each of linear acceleration, rotation rate and the magnetic field strength. Repeated deployments were carried out along a single section of a supraglacial meltwater channel (n = 55). The general applicability of the cylindrical drifter is field-tested, and the repeatability of the sensor time series is determined. Finally, we investigate the potential of the proposed multi-modal drifter to measure the surface transport velocity along a supraglacial channel flow path, and critically assess the device's performance for glaciological applications. The deployment in supraglacial channels allowed the study of the sensor performance in a controlled environment. This is an important step in the development of a reliable measurement platform for supraglacial, englacial and subglacial studies.

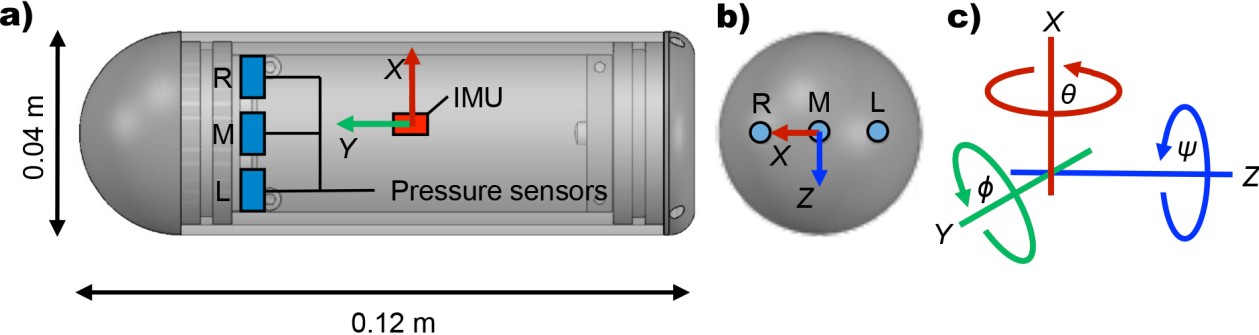

**Figure 1.** Dimensions of the multi-modal drifter used in this work. The drifter includes three identical pressure transducers as well as a single inertial measurement unit (IMU). a): Side-view of the drifter. b): Top-view facing the cap showing the left (L), middle (M) and right (R) pressure ports. c): Body-oriented drifter coordinate system including the roll ($\phi$), pitch ($\theta$) and yaw ($\psi$) angles.

## 2 Methods

### 2.1 Multi-modal drifter

The drifter platform used in this study has two custom machined Polyoxymethylene (POM) plastic end caps and a 4 cm outer diameter polycarbonate plastic tube. The device has a total length of 12 cm, and mass of 143 g. Neutral buoyancy of the drifter

5 is achieved by manually adjusting the length of the sensor by screwing the flat end cap inwards or outwards to increase or decrease the total volume. Small balloons can additionally be attached to the drifter, to fine tune the buoyancy in the field and were used during the field deployment of this study. Each hemispherical end cap of the drifters contains three identical digital total pressure transducers. When the device is submerged in flowing water, the total pressure is the sum of the atmospheric, hydrostatic and hydrodynamic pressures. The devices are designed for a maximum pressure of 2 bar (MS5837-2BA, TE Con-

10 nectivity, Switzerland) and a sensitivity of 0.02 mbar (0.2 mm water column). The pressure sensor data were recorded with a resolution of 0.01 mbar. The accuracy of the pressure transducers was found to be 1 mbar. This was determined by testing fully assembled drifters in a laboratory barochamber up to an equivalent of 55 m of water depth, which is 2.75 times larger than the maximum rated pressure of the sensor. Therefore, the main limitation of the drifter platform results from the measurement range of the chosen pressure sensors, rather than the ability of the mechanical components. Each pressure transducer is

15 equipped with its own on-chip temperature sensor, allowing for all pressure readings to include real-time temperature correction using a two-stage correction algorithm. The algorithm first takes into account device-specific correction coefficients, specified by the manufacturer. In a second step, the device's temperature is used to output the corrected total pressure reading depending on the temperature range. The algorithm is provided on page 7 of the manufacturer's data sheet (TE connectivity sensors, 2017).

All drifters are programmed for atmospheric auto-calibration. Once the device is activated using a magnetic switch, data from each pressure transducer is logged for 15 s. The atmospheric pressure, including any sensor-specific offset, is recorded internally. Afterwards, all three transducers are set to a default value of 100 kPa at local atmosphere. All sensors are therefore auto-calibrated to local changes in atmospheric pressure which occur during the day, directly before each field deployment. This feature removes the necessity of manually correcting pressure sensor readings in post-processing. The drifter units use three pressure sensors (marked as Left, Middle and Right in Figure 1), and can be outfitted with either 2 bar or 30 bar sensors. The drifters were designed this way to include triple modular redundancy by including a pressure sensor array in lieu of a single pressure sensor. The middle pressure sensor (30 bar sensor) was however not used in this study due to the lower sensitivity and range of pressures experienced during channel passage. All following work will therefore only refer to the two lateral (left and right, 2 bar) pressure sensors.

In addition to the three pressure transducers, the drifter platform also contains a digital 9 degree of freedom (DOF) inertial measurement unit (IMU) (BNO055, Bosch Sensortec, Germany) integrating linear accelerometer, gyroscope and magnetometer sensors. The device uses proprietary (Bosch Sensortec) sensor fusion algorithms to combine the linear accelerometer, gyroscope and magnetometer readings into the body-oriented Euler angles to provide real-time absolute orientation at 100 Hz. These IMU sensors were chosen as they represent the current state-of-the-art in IMU technology. Additionally, they have the further benefit that the real-time calibration status of each of the three sensors is recorded (0, lowest to 3, highest) as part of each data set in order to provide quality control information for all IMU measurement data. When running in sensor fusion mode, all variables are saved at 100 Hz, with the exception of the magnetometer, which is recorded at a maximum rate of 20 Hz. The sensor fusion mode of the BNO055 has the major benefit that the absolute orientation, consisting of roll, pitch and yaw angles, is calculated in real-time. The major downside is however that the calibration and sensor fusion used in this procedure are a "black box", as Bosch has not released the algorithms. Previous studies in highly dynamics environments report the measurement error of the BNO055 in the pitch and yaw angles for rapid body movements during driving as less than $0.4°$ and less than $2°$ for the roll angle (Zhao et al., 2017). Similar results were observed for a series of static and dynamic tests using a hexahedron turntable with the BNO055, where the error ranged from $0.53°$ to $0.86°$ for pitch angles, $1.28°$ to $3.53°$ for yaw angles, and $0.44°$ to $1.41°$ for the roll angle (Lin et al., 2017).

The drifters use the STM32L496 microcontroller unit (MCU). They are programmed with the STM32CubeMX software in DFU mode over a USB full speed interface. A ESP8266 WiFi module is used for communication and connected to the MCU via USART2. The IMU and pressure sensors are connected via an $I^2C$ interface and communicate via the $I^2C$ protocol. The data is stored as a delimited text file at 100 or 250 Hz directly to a 6 or 16 GB microSD card. The IMUs were configured to read out more data in addition to the dynamic linear acceleration (body acceleration due to external forcing only, gravity vector removed) and rate of angular rotation (rate gyro), relative to the x-, y- and z-axes of the sensor, as shown in Figure 1. The additional data include the real-time calculation of the drifter body-orientation (3D Euler angles relative to x-, y-, and z-axes and the angles as Quaternions) as well as the 3D magnetic field vector. The orientation of the vector measurements of

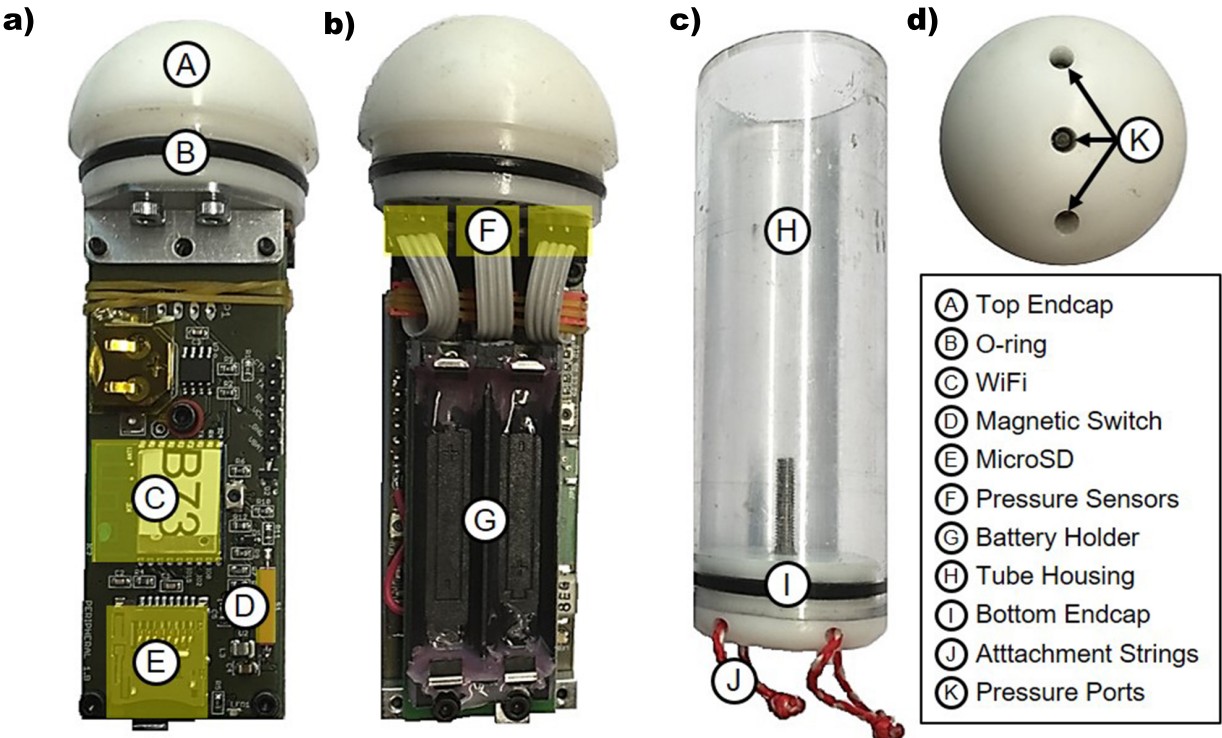

**Figure 2.** Detailed breakdown of the drifter. a): Side-view showing the drifter electronics. b) Side-view showing the reverse side of the electronics board including the battery holder and pressure sensors. c) Polycarbonate tube housing of the drifters with attachment strings for balloons used for manual buoyancy adjustment. d) Top-view facing the cap, showing the ports for each of the three pressure sensors.

the magnetometer readings correspond to the axes of the sensor which are identical to the accelerometer and rate gyro axes. All drifter sensor data is saved as a 27 column ASCII text file. The text files were transferred from the drifters via Wifi to a field computer after drifter recovery from the stream.

5     Vibration and destructive testing of the IMU and the drifter housing have been conducted up to 3000 times the gravitational acceleration, thus showing that the drifter platform can withstand high impacts. Deployment under harsh conditions was successfully proven during measurements inside large-scale hydropower turbines (Kriewitz-Byun et al., 2018). During this study the drifter was only tested in supraglacial streams. Subglacial deployments have however been successfully conducted during subsequent field tests in 2019 and will be described and analyzed in a later study.

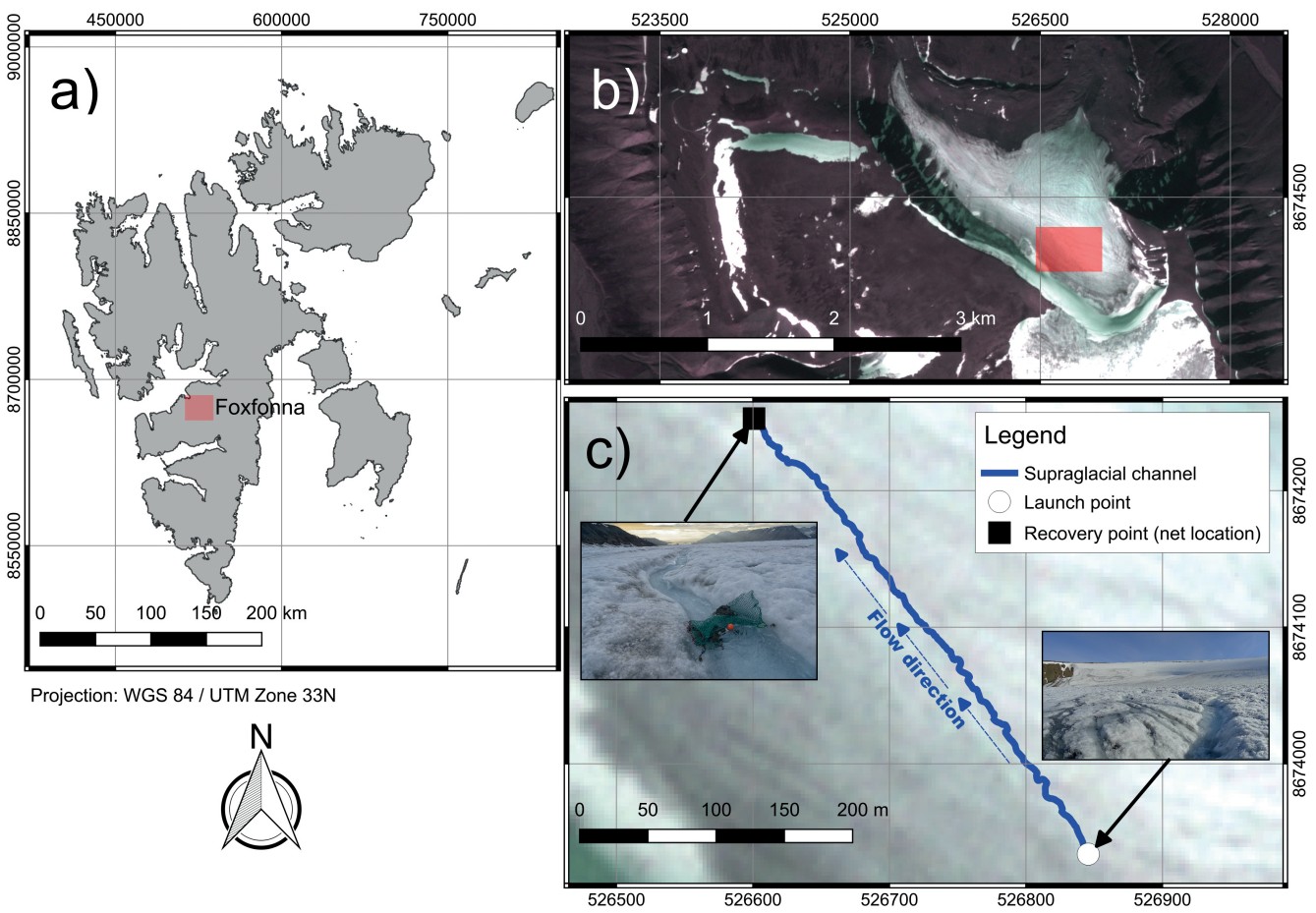

**Figure 3.** a) Location of the Foxfonna glacier on the Svalbard archipelago. b) PlanetScope false color overview of the Foxfonna glacier and the location of the investigated supraglacial channel acquired on 01.08.2018. c) Close up of the studied supraglacial channel. Background image from PlanetScope acquired on 01.08.2018.

## 2.2 Study site

Fieldwork for this study was conducted on the main island of the Norwegian Arctic archipelago Svalbard between the 4th and the 7th of August, 2018 on the approximately 5 km$^2$ big valley glacier Foxfonna. The cold-based, roughly 2.9 kilometers long glacier is located on a northwest-facing slope between 330 and 750 meters elevation above sea level at the end of the Adventdalen valley, next to the main settlement Longyearbyen. The glacier has a network of supraglacial channels developing on the surface of the glacier during the summer ablation period. Some of the channels cut deep enough to form englacial cut-and-closure systems (Gulley et al., 2009) and others remain partially snow-plugged during the summer. Additional channels emerge at the glacier front, indicating existing subglacial drainage channels.

## 2.3 Field Deployments

Two different experiments were conducted on the glacier surface. The first experiment tested the general feasibility of the drifters traveling through an englacial/subglacial system and being recovered. This involved deploying five wooden drifter surrogates identical in size to our multi-modal drifters in a 2.5 km long supraglacial, partly englacial channel. The channel on the eastern side of the glacier had several well-developed step-pool sequences and was incised deeply into the glacial ice. Further downstream, the channel developed into a partially snow-plugged, partially englacial system. The final channel section had a large amount of rock debris typical of subglacial environments. A net was installed where the channel reemerged on the surface at the eastern lateral moraine. Emerging wooden sensor surrogates were trapped in the net and recovered by removing them by hand.

The second, main experiment was conducted along a 450 m section of a supraglacial channel on Foxfonna. The investigated section had a total elevation difference of 30 m (handheld GPS accuracy of 5 m) as measured between the start and the end of the channel section. The section included several step-pool sequences as well as rapids and recirculation zones. The experiments were conducted within this channel section for three main reasons. First, the purpose of the study was to determine field measurement repeatability, requiring a channel with different morphological features. Second, with only five prototypes, the risk of losing a drifter had to be kept low. Finally, the study of the supraglacial system allowed for the filming of deployments, and this provides a simple and robust evaluation method to compare the sensor data with observed movements of the drifter within the flow. All five drifters were launched from the location, marked with a white circle on the map in Figure 3, and were recovered using a marine fishing net, installed at the downstream end of the channel section. A total of 55 drifter deployments with five individual multi-modal drifters were conducted. 10 deployments were collected in the afternoon of the second field day (05.08.2019 15:52 -17:20 local time) and the remaining 45 deployments in the late evening and night of the fourth field day (07.08.2019 18:53 – 23:37 local time). The discharge varied throughout the deployment time, depending largely on weather conditions (sunny, with increased melt on 05.08 and cloudy, rainy, and sunny on 07.08). The exact discharge was not recorded. Some of the deployments had slightly varying buoyancy, due to varying balloon inflation, but all deployments on the 450 m section were conducted with a single balloon. Four out of the 55 deployments had the drifters connected in tandem with cable ties, to test the variation between the sensor readings of two different drifters passing through nearly identical flow paths. All of the drifters were switched on and then left on the ground for at least 30 s before deployment and for an additional 30 s after successful recovery from the stream and before switching off. This was done to ensure that the drifters had enough time for self-calibration to atmospheric pressure before the deployment and the IMU sensor readings could calibrate and provide constant-value readings, which later serve to mark the start and stop of each deployment.

## 2.4 Data preparation and processing workflow

All data processing was performed using Matlab R2018b. Corrupted datasets with missing data or faulty sensor readings were removed (n=9). In cases where a drifter switched off and back on again during a deployment, multiple files were concatenated into a single dataset. The start and end of each dataset were manually trimmed such that the processed time series only represent the time within the glacial stream. Threshold criteria used, for determining the start of measurement, were the linear acceleration peaks of a drifter's first impact with the water surface during deployment as well as the final impact when the drifter contacted the net during recovery. Entries in the dataset with no data or poor calibration status were filtered out in the next step. After trimming, the time series data were filtered for outliers with the following thresholds: $\pm200\,\mu\text{T}$ for the magnetometer, $\pm60\,\text{m s}^{-2}$ for the linear accelerometer and $\pm50\,\text{deg s}^{-1}$ for the rate gyro.

We defined a recovery as well as a utility rate for the drifter deployments to assess their overall performance. The recovery rate describes the rate of recovered drifters for each set of deployment. The data usability rate gives an indication for the amount of usable datasets for all recovered drifters. This is important, as not all recovered datasets are usable due to technical problems. The overall utility rate allows for an estimation of the total amount of needed drifter deployments based on their recovery and data usability rates. Thus being a valuable tool for fieldwork planning.

$$\text{Recovery rate} = \frac{\text{Number of recovered dummies/ drifters}}{\text{Number of deployed dummies /drifters}} \tag{1}$$

$$\text{Data usability rate} = \frac{\text{Number of usable datasets}}{\text{Number of recovered drifters}} \tag{2}$$

$$\text{Utility rate} = \text{Recovery rate} \cdot \text{Data usability rate} \tag{3}$$

The statistical analysis evaluated the degree of agreement between individual sensor time series for each deployment to assess the repeatability of the drifter field data. Pearson product-moment correlation coefficients were used to investigate the correlation structure between different sensor modalities, and to assess if the different modalities were dependent or independent variables. The associations were classified with a modified scheme from Cohen (1992). In the next step, the empirical probability distributions were investigated together with the statistical moments mean, variance, standard deviation, skewness and kurtosis. Afterwards, the empirical probability distributions of each deployment were compared to the ensemble empirical probability distributions to determine the measurement repeatability. To ensure a robust assessment of repeatability, several criteria were evaluated: chi square distances, mean absolute error, mean squared error, data ranged normalized root mean square and the Kullback Leibler divergence (Kullback and Leibler, 1951).

The assessment of the minimum needed sample size to achieve a given precision of each mode (e.g. total pressure, linear acceleration in the x – direction) was done following the equation from Hou et al. (2018)

$$n = \frac{Z_{1-\frac{\alpha}{2}}^2}{\varepsilon^2} \cdot \left(\frac{\sigma}{\mu}\right)^2 \tag{4}$$

where $Z_{1-\frac{\alpha}{2}}^2$ is the standard normal deviate (e.g. $Z_{0.975} = 1.96$ for the 95% confidence interval), $\varepsilon$ is the defined precision, $\sigma$ is the standard deviation of the population and $\mu$ is the population mean. In this study, we set the desired error of the sample mean to be within ±10% of the true value (i.e. $\varepsilon = 0.10$), 95% of the time (i.e. $Z_{0.975} = 1.96$). The values for $\sigma$ and $\mu$ were then obtained from the statistical analysis of the time series data from all deployments.

To find potential features in the time-series and to test the degree to which data vary over time, moving means were calculated over the dataset. The moving means for this study were calculated over a time window of 5 s, as potential signal features were most prominent at this window length, and plotted together with the 95% confidence interval (CI). This was done for 40 of the deployments for the first 200 s of the channel passage. The analysis was limited to the first 200 s, as not all drifters recorded for the full length of time, so that a compromise had to be found between maximizing the total number of deployments as well as the total duration of deployment. The other 15 deployments out of the total 55 deployments were left out as they either recorded no data (n=9) or recorded only for parts of the passage (n=6) leading to very short datasets.

The surface transport speed was calculated by integrating the acceleration measurements over a rolling time window for the remaining 40 deployments (n=40). The window width was initially randomly chosen for the velocity calculation. Once the three components of the velocity vector were calculated, we defined the transport speed as the magnitude of the velocity. The transport speed was then compared to the estimated transport velocity, which was found by dividing the transport distance (450 m) by the total travel time of the drifter. The integration window size was then readjusted individually for every deployment, so that it would be within a 10% error threshold from the drifter's estimated transport velocity. By doing so, the individual changes in the observed transport velocity are accounted for. As the acceleration includes rapid changes in rigid body motion, for instance due to impact with the channel walls, we found that integration produced large outliers, which are not representative of the water flow itself. Therefore the estimated instantaneous velocities, exceeding $10\,\mathrm{m\,s^{-1}}$, were removed.

## 3 Results

### 3.1 Utility rate

In the first experiment, five wooden dummies, of the same size and buoyancy as the drifter, were deployed in a 2.5 km long supraglacial, partly englacial channel with features of subglacial channels (step-pool, glide and chute sequences and debris at the channel bootom) , and four out of five dummies were recovered after 72 hours. The second experiment consisted of 55 multi-modal drifter deployments in a 450 m supgraglacial channel section, returning a total of 40 useful datasets. The other

**Table 1.** Recovery and utility rates for supraglacial, as well as englacial/ subglacial deployments. The results from the supraglacial system are from the second experiment with 5 drifters and a total of 55 deployments in the 450 m long supraglacial channel section. Subglacial/englacial system rates are based on the assumption that the drifters can pass through the system, if the dummies pass through. The Estimated total utility rate for subglacial/ englacial deployments is based on the dummy deployment, as well as the data usability rate.

|  | Supraglacial | Englacial/ Subglacial |
| --- | --- | --- |
| Recovery rate | 1.00 | 0.80 |
| Utility rate | 0.73 | 0.58 |

15 deployments had datasets of insufficient duration (below 200 s, compared to an average transit time of 360 s), as drifters only recorded part of the deployments there (n=6) or recorded no data at all (n=9). The results for recovery and utility rate are presented in Table 1.

## 3.2 Statistical evaluation

We calculated the ensemble statistics for all successful deployments and numerical values can be found in the supplementary material (Table S1 and S2). The mean values and the standard deviations were then used to estimate the required sample size to achieve a precision of the sample mean to be within ±10% of the time-averages (i.e. $\varepsilon = 0.10$) for 95% of the time (i.e.$Z_{0.975} = 1.96$). The obtained sample size estimates were afterwards multiplied with the utility rates to estimate the required number of supraglacial and subglacial/ englacial deployments. The mean pressure values were thereby corrected with the calculated air pressure of 941.8 hPa based on elevation (600 m) and air temperature on 07.08.2018. This calculation resulted in unrealistically high required sample sizes (Table 2). These high numbers are however composed of several components: One part is caused by the sensor accuracy and technical problems causing high variations in the measured data, the second part of the inaccuracy is due to spatial and temporal flow variability both between deployments, but also along the flow path. The lowest required sample size was for the pressure sensors and the magnetic field intensity magnitude. The latter should however also be corrected by the value of the local magnetic field strength and the number of required deployments is therefore likely to be higher.

Distance and similarity measures were used to test the repeatability of the datasets. All calculated values for every sensor modality and statistical measure (Chi Squared Error, Kullback Leiber divergence, mean average error, mean squared error and data ranged normalized root mean square) are close to zero, thus indicating a high repeatability of the drifter deployments (Table S2 in the supplement). Our calculations of the Pearson correlation coefficients confirm, that the two pressure sensors are redundant (Figure 4). Additionally there is a correlation between the pressure sensors and the Magnetometer Y readings. The other sensor modalities represent independent variables.

**Table 2.** Estimated multi-modal sample sizes for ±10% precision and a 95% CI based on measured mean values and standard deviations from all deployments (n=40), as well as estimated sample sizes for supraglacial and subglacial deployments based on the utility rate and the measured mean values and standard deviations.

| Sensor mode | Required sample size estimate | Supraglacial | Subglacial |
| --- | --- | --- | --- |
| Pressure left | 2 | 3 | 4 |
| Pressure right | 2 | 3 | 4 |
| Magnetometer X | 1264 | 1732 | 2180 |
| Magnetometer Y | 531 | 728 | 916 |
| Magnetometer Z | 296 | 406 | 511 |
| ‖Magnetometer‖ | 3 | 4 | 5 |
| Accelerometer X | 479,603 | 656,991 | 826,902 |
| Accelerometer Y | 7259 | 9944 | 12,516 |
| Accelerometer Z | 1,382,976 | 1,894,488 | 2,384,442 |
| ‖Accelerometer‖ | 670 | 918 | 1155 |
| Gyroscope X | 115,419 | 158,109 | 198,999 |
| Gyroscope Y | 14,309,576 | 19,602,159 | 24,671,683 |
| Gyroscope Z | 301,182 | 412,578 | 519,280 |
| ‖Gyroscope‖ | 281 | 385 | 485 |

### 3.3 Moving mean analysis and velocities

After filtering out all short-term sensor fluctuations with a 5 s rolling time window a clear redundancy between the pressure sensors (Figure 5), as well as a close correlation between Magnetometer Y and pressure readings (Figure 6) become visible. As the experiments are conducted at atmospheric pressure conditions with only small elevation change over the passage, almost homogenous pressure signals should be expected. The plot in Figure 5 however, shows that the pressure records are displaying distinct variations including sharp peaks, sudden increases and drops. These variations are superimposed on a general increase of the pressure, which might be caused by the increasing atmospheric pressures as the drifters are flowing downhill, as well as increasing water depths in the channel. The 95% CI of the averaged pressure signal can be seen to vary over time, but generally follows the same features as the average, with some features having smaller CIs than others. The values of the 95% CIs are generally very low and on average do not exceed values above ±0.11% of the mean pressure value of 1005 hPa.

Plotting the magnitudes of the different sensor modalities shows, that the obtained signals are not homogeneous over time, but rather have pronounced signal variations, as also visible in the pressure signals. The width of the CIs of all sensor modalities decrease after drifter deployment, vary however slightly throughout the time series. This is due to the drifters passing through different channel geometries and flow features at individual velocities. The magnitude of the magnetometer has the

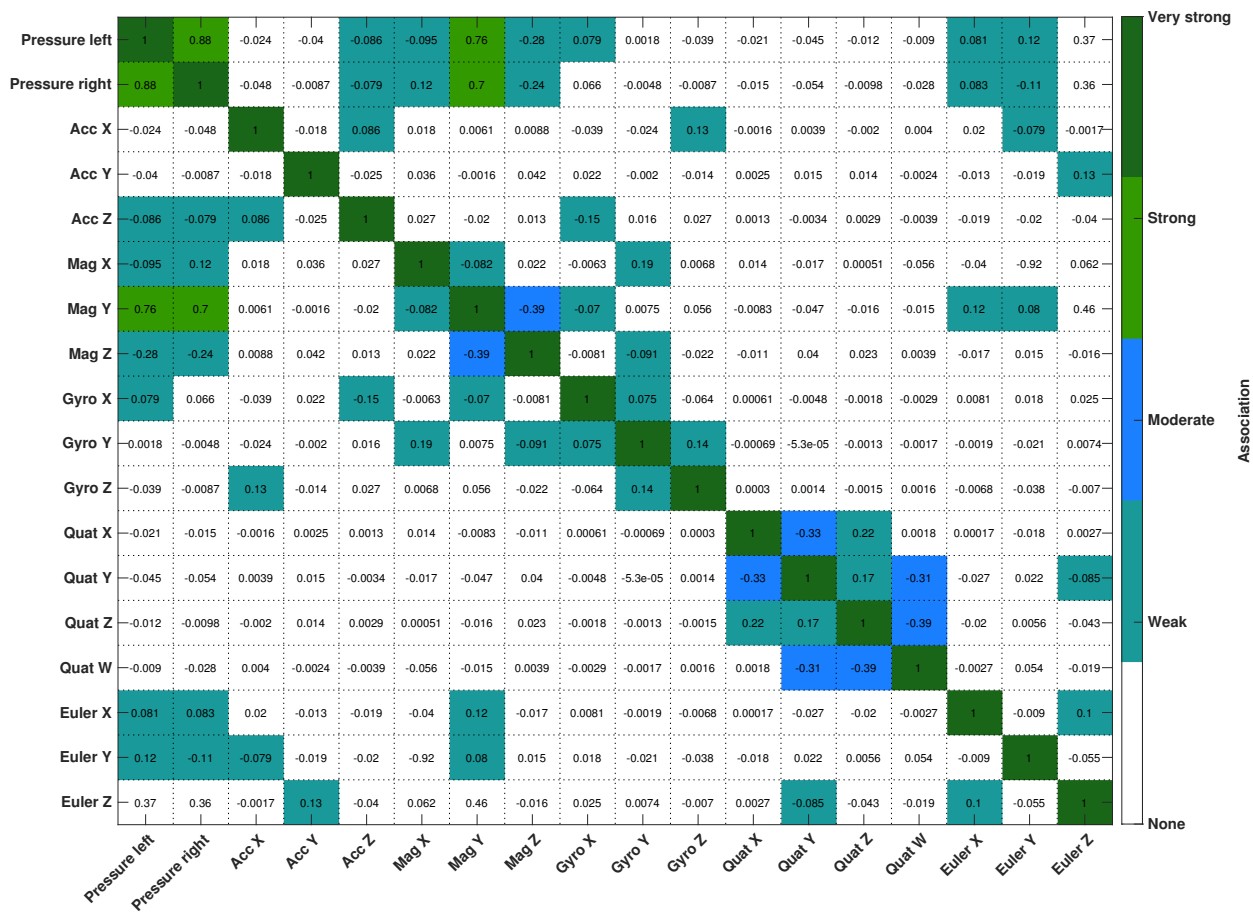

**Figure 4.** Pearson product-moment correlation coefficients (r) between the different sensor readings for all deployments (n=40). The classification is adapted after Cohen (1992).

second smallest 95% CI with a mean CI of ±2.45% of its' mean value of 54.6 µT. The other sensor modalities have larger CIs with gyroscope readings being the next lowest on the list with a mean CI of ±24.8% of it's mean value of 3.8 deg s$^{-1}$. The accelerometer has the largest CI, and hence largest variation of recorded values, with a mean CI of ±34.4% of its' mean value of 2.54 m s$^{-1}$.

5     The velocities in the x-direction (sideways in the plane of the drifters longitudinal direction, see also Figure 1) alternate between positive and negative values as the drifters travel through a meandering channel (Figure 8). Velocities in the y-direction remained mostly negative and vary between fast and slow zones. The negative values in y-direction are due to the hydrodynamics of the drifters and the balloon, which together with the currents lead the drifter to face upstream. Every drifter had one balloon attached to achieve neutral buoyancy. We observed that the currents lead to the balloon flowing slightly ahead and

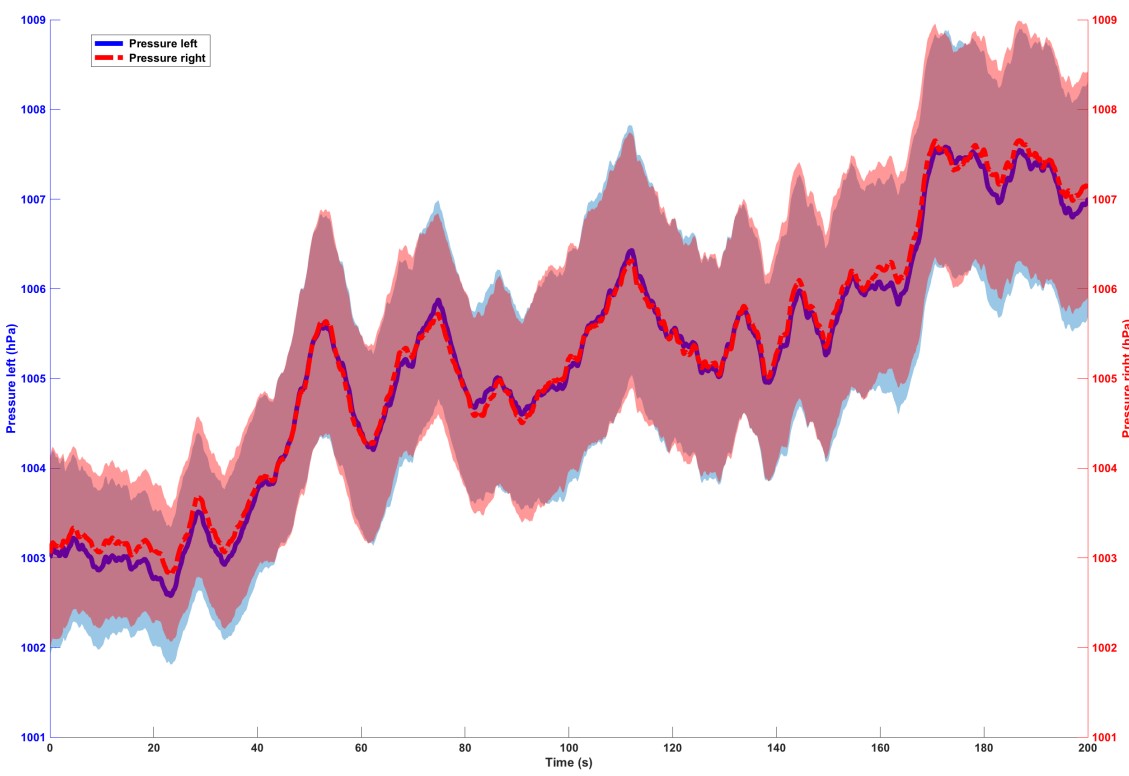

**Figure 5.** Mean values and 95% CI (shaded area) of the left and right pressure over 40 deployments (n=40) over the first 200 seconds of the flow path passage. The data is averaged over a 5 s time window and across 40 deployments.

the drifter facing away most of the time, thus leading to negative accelerations and velocities in the y-direction (longitudinal direction of the drifter, see Figure 1). In the z-direction (downwards facing from the drifters longitudinal plane, see Figure 1) the velocities remained mainly positive and vary between zones with slower and faster flow. The magnitude of the velocity shows several pronounced signal variations in the time series as well. Generally values around $2\,\mathrm{m\,s^{-1}}$ are most common. The mean value of the 40 deployments, used for acceleration integration, is $1.94\,\mathrm{m\,s^{-1}}$ and the mean 95% CI is $\pm24.5\%$ (n=40).

### 3.4 Signal features of a step-pool sequence

Video footage was taken periodically during the deployment and allowed to isolate the acceleration and pressure record during passage of a small step-pool sequence (Figure 9). A pronounced peak, followed by a drop in the signal for the chosen time period, where the drifter passed over the step-pool sequence is visible. The pressure signal trails behind the acceleration signal.

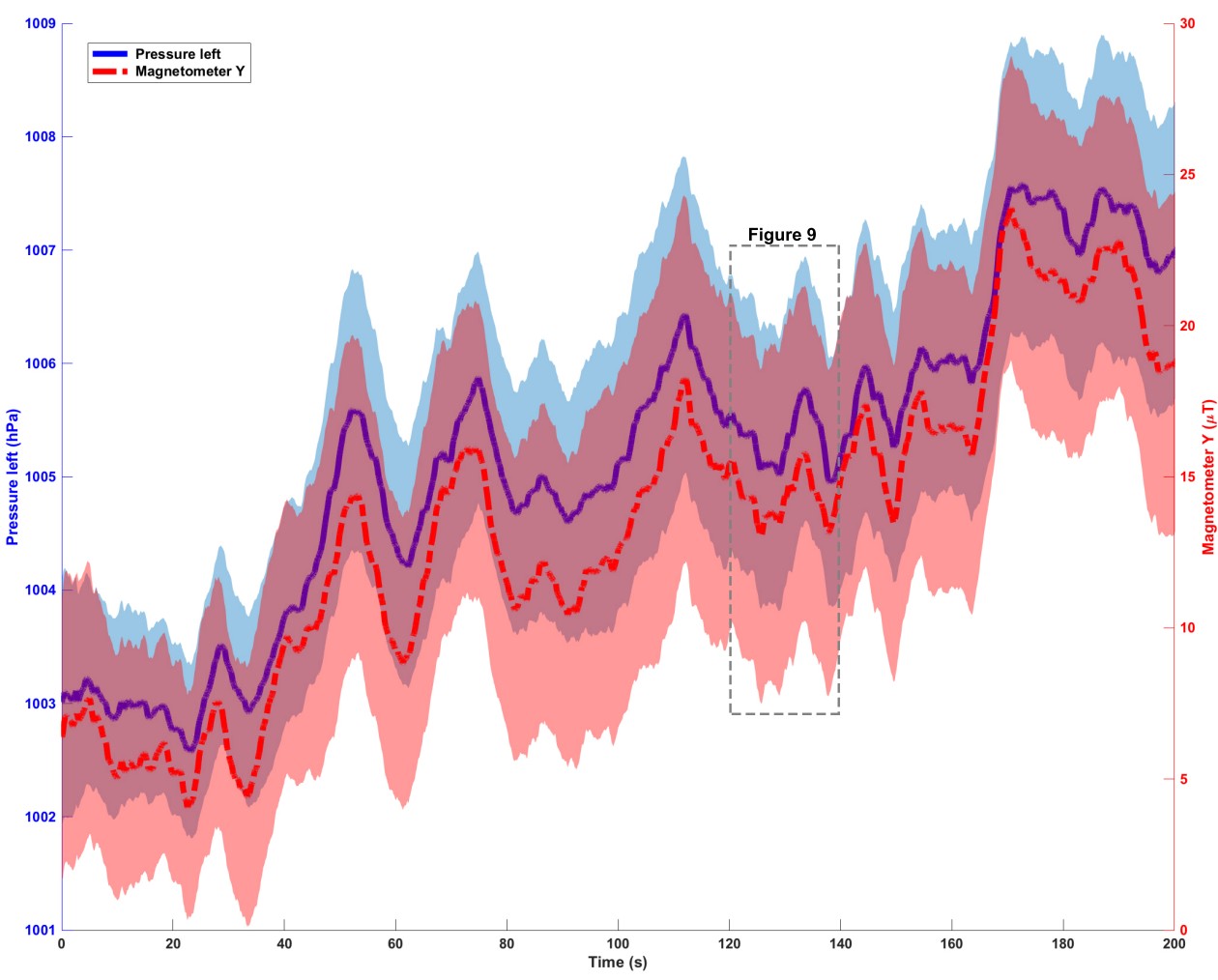

**Figure 6.** Left pressure and magnetometer in y-direction time series. The plot shows the first 200 seconds of 40 deployments (n=40), with a moving mean with a time window of 5 s and the 95% CI (shaded area).

The video footage shows that the drifter was speeding up towards the edge of the step, when the pressure and the acceleration signal increased (Figure 9, A)). As the drifter flowed over the edge and dropped into the pool underneath, the pressure and acceleration dropped. Once in the pool, the drifter was caught in a recirculating current and remained in the pool for several seconds. This leads to a drop of the signals, which stagnate at a lower level before increasing again, once the drifter leaves the pool and flows onward in the supraglacial channel.

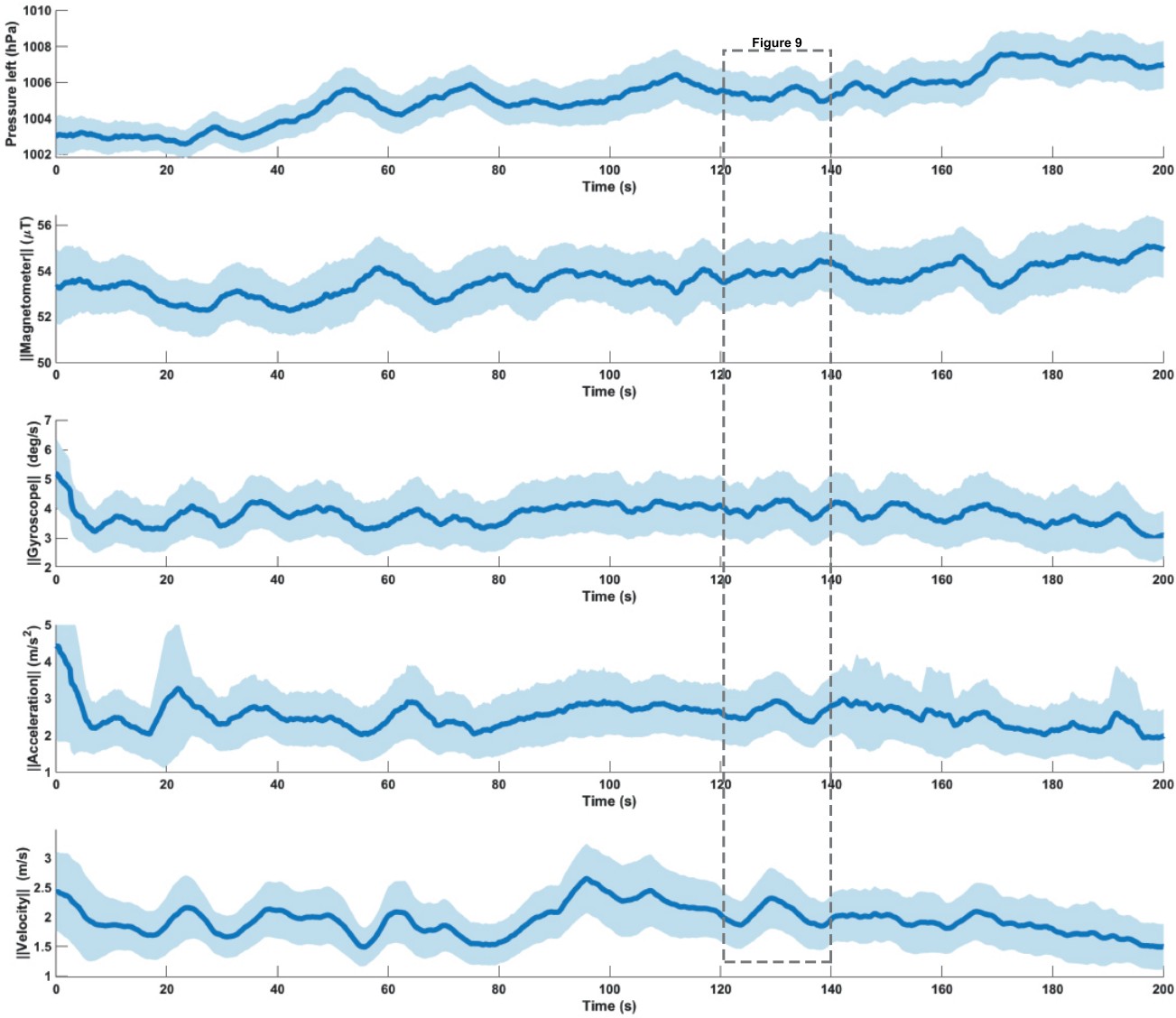

**Figure 7.** Left pressure overlayed with the magnitudes of the magnetometer, the gyroscope, the accelerometer and the velocities, obtained from acceleration integration. The line is the moving mean with a 5 s time window and the shaded area represents the 95% CI of 40 deployments (n=40).

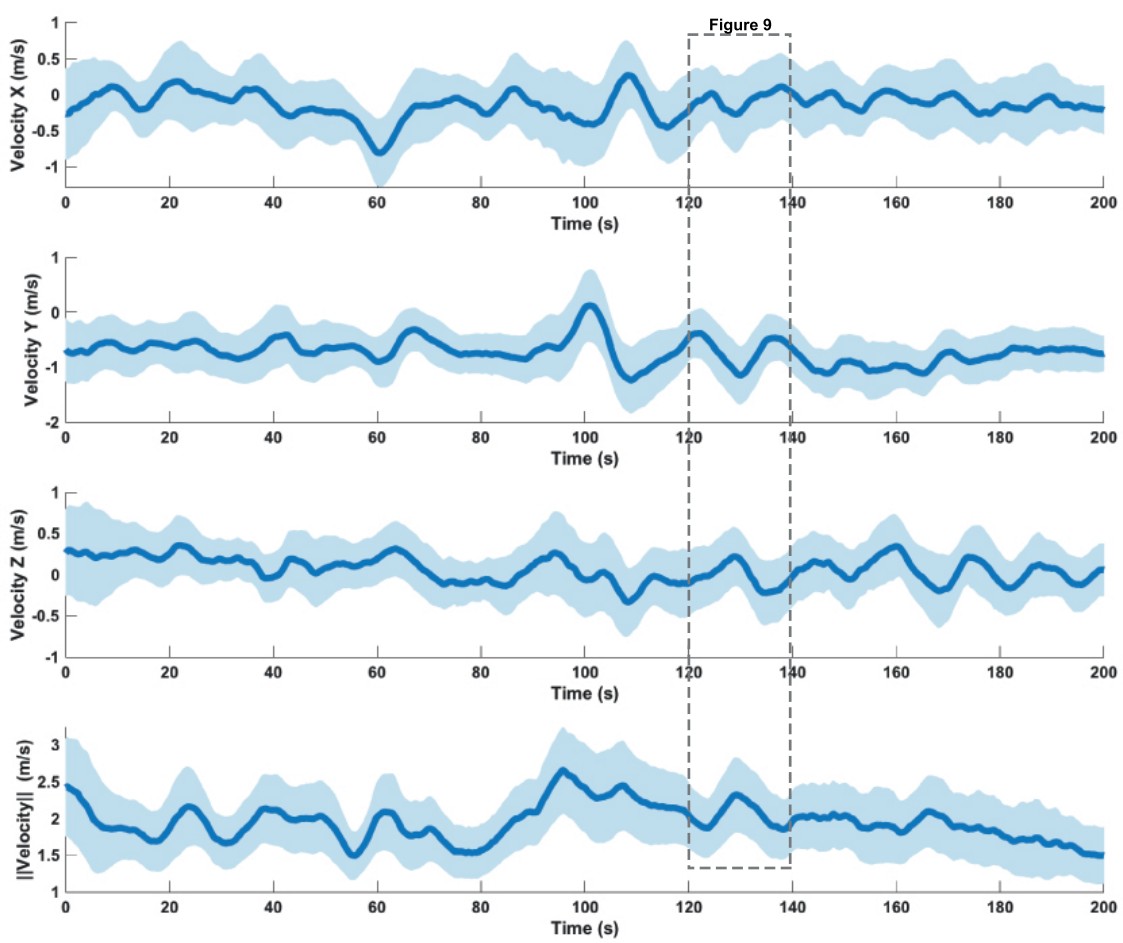

**Figure 8.** Mean values and 95% CI of the three velocity components and velocity magnitude of 40 deployments (n=40) over the first 200 seconds of the flow path passage. The data is averaged over a 5 s time window.

## 4    Discussion

### 4.1    Drifter performance

An investigation of a glacial stream using sensing drifters generally demands a significant number of deployments (Table 2), to allow for a statistical analysis and to account for drifter loss and technical problems, which are expressed by the calculation of
5    the utility rate. Pressure values are the easiest to acquire, as they need the lowest number of deployments compared to the much higher deployment numbers for the IMU values. However, the acquisition of flow data with sensing drifters in glacial channels

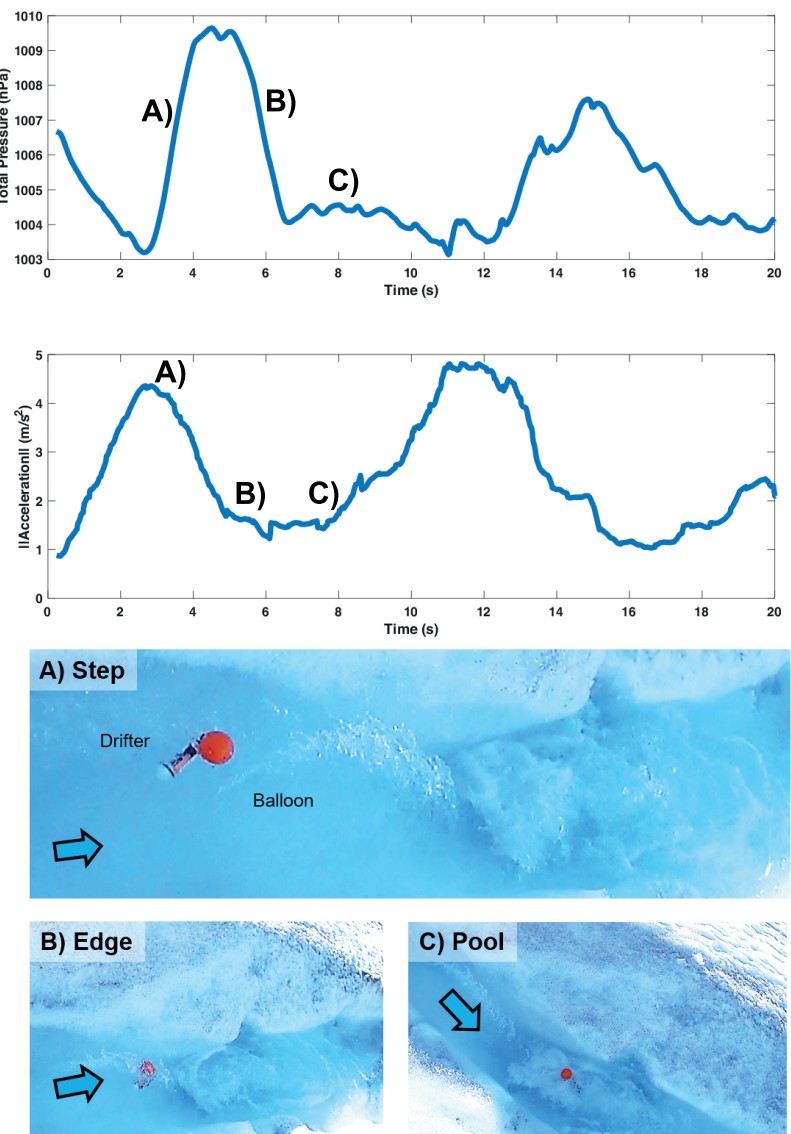

**Figure 9.** Exemplary data of a drifter going over a step-pool sequence. The plot shows the left pressure record as well as the magnitude of the acceleration from a single drifter, while passing over a step-pool sequence. The data is averaged over a 2.5 s time window. Three zones are marked on the plot, which represent different parts of the passage. A) The acceleration and the total pressure increase, as the drifter travels towards the edge of the step. B) The acceleration and the total pressure drop as the drifter flows over the edge. C) Nearly constant acceleration and pressure values while the drifter is caught in an eddy inside the pool.

will require an unrealistic amount of time in the field, and the deployment of many drifters simultaneously to reduce field time and the potential for external factors to influence the measurements (e.g. discharge variations). Bagshaw et al. (2012) previously showed that drifter passage through glacial channels can take several days to weeks, which imposes a practical challenge when it comes to acquiring several hundred to several thousand deployments for statistical analysis. In practice, this means that the measurements with sensing drifters will only be possible with lower statistical significance (p » 0.05), as field deployments of several thousand drifters are not realistic. The number of deployments can however be reduced by decreasing the acceptable error, which is introduced by the sensor accuracy and technical problems, as well as through improved field deployment and recovery methods. Further technological improvements of the sensor platform could reduce the signal to noise ratio of the drifters and the data usability rate. A problem with the proposed drifter platform was for example an occasional loss of battery contact in the battery holder due to high impacts, leading to corrupted data. This problem has been subsequently solved in an updated drifter system, which was field tested in summer 2019. The recovery during the presented field tests was done by the installation of a net inside the glacial stream. A method, that works well inside smaller supraglacial streams. It bears however the challenge of high ice (supraglacial streams) and bed loads (glacier outlets) clogging the net. This leads to the net damming up the water and hence allowing the drifter to flow over the net, requiring a regular maintenance of the net to prevent drifter loss. High discharge and flow velocities pose additional challenges and therefore require the further development of recovery methods.

The analysis of the moving means of the signals show that clear features become recognizable in some of the signals over the channel passage. The analysis of videos from the deployments show that these patterns seem to be related to geometrical features in the flow path such as step-pool sequences and recirculation zones. We did however only record videos from thirteen deployments and did not measure channel geometries during this field experiment. More field studies with known geometry and repeated measurements of geometric features, which can be detected and classified in the signal time series and the channel geometry are therefore required to verify this hypothesis. Pressure sensors and magnetometer Y seem to produce the most clear signal features. The pressure sensors have also the smallest 95% CI of ±0.11% relative to the mean, and deliver the most repeatable data with the lowest error. The CIs for all sensors remain more or less constant over time, some CIs are however larger than others as the sensors travel with different velocities through the channel and pass certain geometric features at different times, which is not accounted for in the plot over time. A part of the higher CIs also comes from the individual drifter movements, such as rotation rate, which are different for every deployment, hence leading to a higher CI. It is however still possible to get pronounced signal features of the IMU readings as well, which can then be linked to geometric features of the channel and flow morphology, as shown in the supplementary video sequences. The IMU readings hereby give an extra value compared to a platform only equipped with pressure sensors, as they can provide the necessary extra information, which will allow us to further distinguish between different geometrical and morphological features of the flow and the channel respectively.

Multi-modal drifters with inertial measurement units present a potentially valuable tool to obtain three-dimensional accelerations along the flow path of a glacial channel. The integration of this data allows to obtain three-dimensional velocity estimates

along the channel and to obtain transport surface velocities. This allows for an initial, first order of magnitude estimate of the large-scale (> 10 cm) velocity distribution inside glacier channels, offering a large improvement compared to the state of the art, which relies on point velocities at certain locations through boreholes or integrated velocities along the flow path obtained from dye tracing. The velocities obtained from acceleration integration should however be further constrained with field measurements in future studies to deduce the error introduced by the integration. Nevertheless, the average 95% CI of ±24.5% relative to the mean value clearly implies that further improvements are in order.

It can generally be stated that the proposed multi-modal drifter platform provides repeatable data considering the supraglacial field experiments at Foxfonna. The utility rate of 73% in supraglacial channels and 58% of the total deployments in englacial/subglacial channels provides a first, reasonable estimate of how many sensors a practitioner should consider deploying.

## 4.2 Glaciological implications

This study establishes that multimodal sensing drifters equipped with pressure sensors and an inertial measurement unit present a new tool to obtain repeatable measurements in supraglacial channels. Further field studies are needed to interpret sensor time series to identify specific features corresponding to channel morphological types and flow conditions. The resulting signal features may be used to provide new insights into the dynamics of glacial hydraulics by overcoming the limitations of existing technologies, which are typically either restricted to a point location or yield only information integrated over the flow path.

We believe that multi-modal sensing drifters can also be of great value for the modeling community by providing input for various models, like subglacial hydrology (e.g. Werder et al., 2013) or supraglacial channel development (e.g. Decaux et al., 2019). However, additional fieldwork using ground truth velocities from established measurements (e.g. tracer studies) as compared to drifter estimates are necessary. Once this is done, other important studies linking subglacial hydrology measurements to glacier dynamics can be envisaged.

## 5 Conclusions

The multi-modal drifter platform tested in this work measures the total water pressure, linear acceleration, magnetic field strength and rotation rate while flowing along a glacial channel. The field experiments in this study showed that the platform used is able to obtain repeatable data in a 450 m supraglacial stream section. The multi-modal drifter measurements appear however to require a significant number of repeated deployments to yield repeatable statistics at a 95% CI. This is due to a combination of technical problems and deployment losses as well as natural flow variability. Rapid changes in channel flows will always cause the recorded signals to have some variations between deployments. We show that it is possible to estimate the number of deployments as a percentage of a given sensor mode's time-averaged value. It was observed that increasing

the error threshold to above 10% of the time-average can significantly reduce the number of necessary deployments. The total pressure measurement was found to be the most feasible for repeatable flow path measurements in supraglacial channels, as they consistently had the lowest error thresholds and high repeatability. After integration and low-pass filtering, the linear acceleration allowed for an estimation of flow velocities. An interesting finding was that the drifter data do not have random

distributions, but rather distinctly non-Gaussian probability distributions. Comparison of time-series events with video footage of the drifters indicates that rapid variations in the drifter data likely correspond to changes in the channel morphology (e.g. step-pool sequence) and their corresponding flow characteristics (e.g. turbulent jet or recirculation region). We are optimistic that linking distinct signal variations to channel morphology and flow properties may provide further insights into unknown channel geometries, e.g. in subglacial channels. This additional information may provide new, more efficient means to inves-

tigate the velocity distributions within glacial channels. Future field studies including distributed velocity mapping will be carried out to further the technological improvements of the proposed platform, with the long-term objective of providing a new robust, reliable and affordable device for glacial hydrological studies.

*Video supplement.* Sample videos of the drifter deployments in supraglacial channels can be found online as video supplements on the journal webpage.

*Author contributions.* AA, MK and JAT designed and planed the study. AA and MK conducted the fieldwork with support of AJH. AA analyzed the data and wrote the manuscript. All authors contributed to the interpretation of the results and the manuscript.

*Competing interests.* The authors declare that they have no conflict of interest.

*Data availability.* All raw data is available under doi: 10.5281/zenodo.3660488.

*Acknowledgements.* AA and AK acknowledge partial support by the ERC project ICEMASS. The transport of AA to and from Longyearbyen

was provided by Oceanwide Expeditions. Development of the drifter platform and the field costs of MK were paid by the Horizon 2020 FITHydro project as well as the Research Council of Norway through the Centre of Excellence funding scheme, project number 223254 – NTNU AMOS and the Estonian Research Council grants IUT-339 and PUT-1690 Bioinspired Flow Sensing. The University Centre of Svalbard provided logistical support and the TalTECH IT doctoral school provided support for a research stay of AA in Tallinn for data analyses. This study is a contribution to the Svalbard Integrated Arctic Earth Observing System (SIOS). The reviews of Liz Bagshaw and

Samuel Doyle helped to improve the quality of the manuscript. We would further on like to thank Samuel Doyle for his second review as well as Jan De Rydt for his Editorial work.

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
