# Peer review of "Pressure and inertia sensing drifters for glacial hydrology flow path measurements"

_The Cryosphere, 2019_

## Referee Comment (RC1) · Elizabeth Bagshaw (Referee) · 16 Aug 2019

This paper gives a comprehensive account of a challenging experiment to assess the repeatability and usability of data collected by a drifting sensor. It should be noted that experiments including field testing of new engineering techniques in extreme environments are exceptionally challenging, and the authors should be congratulated on their successful deployment. The paper is well-written, and gives a comprehensive review of much of the supporting literature.

However, the scope of the paper is somewhat confusing to the reader, since the problem is framed as a subglacial experiment, yet the data are confined to the supraglacial

environment. This in no way invalidates the results, but I believe that the paper would be more appealing to the target audience if the supraglacial scope were made clear from the first paragraph and in the abstract. The subglacial deployment may be the ultimate goal of the project, but the current state of the science and engineering is limited to the supraglacial. I advise that the introductory paragraphs and abstract are focussed on supraglacial literature, with some additional references as necessary (for example, Decaux et al 2019 cited later). The subglacial material can then be moved to later in the discussion, to make clear how the supraglacial results can be utilised and developed in the future.

I like the use of statistics to validate the sensor performance, and the realism in relating the statistical results to logistical practicality. However, the actual purpose of the paper is not entirely clear in this iteration – is it an engineering test, a sensor validation exercise, or does it reveal a previously unknown glaciological phenomenon? All of these are valid outcomes, but the introduction and particularly the abstract should be better focused to demonstrate that experimental purpose to the reader. The paper could also be shortened by moving some of the tables to supplementary info (see below).

Specific comments Figure 1: please could you include a labelled photo of the sensors? It would be great to see them in a bit more detail Figure 2: Place names are illegible on the map, and the features of the glacier need some labels in c). It would be useful to know where the net was situated, for example, and perhaps have some accompanying photos of the deployment/recovery sites Could you comment on the feasibility of the net recovery in a larger system, and if the sensors are destined for the subglacial system, on the robustness of the net methods with debris and bedload transport in the flow? Table 1: Could transmission distance (if relevant) be a separate column rather than a comment? P3, L23: Is 500 Euro truly low cost?! This is a subjective term. P10, L20: Please include an estimate of the range of discharge variability Figure 3 doesn't really add much to the paper, it could be removed to save space without detriment, since the workflow is not unusual and is described in the text P12, L25: please define the

'features of subglacial channels' Figure 4 and Tables 3 and 5 may be better placed in supplementary info, since their content is only of interest to a very specific audience and the paper is rather long Table 4 and P14, L7: what is the 'required size' for a subglacial deployment? Unclear how these measurements are extrapolated to the subglacial system: just because the drifter can move through an open supraglacial stream doesn't necessarily infer that it will pass through the subglacial environment Figure 10: Can other sensor data be added to this figure? It would be very useful to see the accelerometer data plotted alongside. The IMU accelerometer method is really exciting, so if the data could be demonstrated alongside the pressure and photographs, it would really contribute something valuable to the field. P25, L11-25: What were you hoping to determine with this dataset? It seems that you have proven that the technology and the sensor set work (which is great!), so can you relate this to the flowpaths? How do the data relate to visual observations? If you hope to use this to visualise subglacial systems, then it is important to relate the sensor data (of which you have a considerable quantity) from the supraglacial system to visual observations where you can. Then you can demonstrate how this might be used in the subglacial environment. 'We need more data' seems a bit of a cop-out! What precision do you need to obtain scientifically useful data? P26, L26: What is 'satisfactory performance'? This is very subjective. What did this experiment hope to achieve, and did you do it? Was it field testing of the casing, the transport method, or of the sensor performance, or of the usefulness of the data to characterise the supraglacial flowpath? Or of future subglacial deployment? Please be specific – this is an excellent engineering test, but subjectivity in appraisal should be avoided. Table 6 isn't terribly useful.

---

## Referee Comment (RC2) · Samuel Doyle (Referee) · 20 Aug 2019

Alexander et al. present a statistical assessment of the performance of a new sensing system – a Lagrangian drifter – for glacier hydrological experiments. They report results from repeated tests in a supraglacial channel and suggest (though never that directly) that there may be future potential for deploying the drifter within the subglacial environment. The sensor system is novel and this manuscript makes an important contribution to the very limited literature on Lagrangian drifters in glaciology. Although it is verbose, the paper is generally well-written. The figures and tables are clear, though the number of tables and figures within the manuscript could be reduced. Citations are

appropriate, however, it is unclear why the introduction focuses (e.g. Table 1) on wireless in situ sensor systems, which are not really relevant, while previously published drifter studies from fluvial and oceanographic studies are not discussed in detail. I have three major comments and several minor comments listed below.

Major comments

1. With the exception of the "Moulin Explorer" and the eTracer, the introduction lacks a section describing what drifters are currently available (or have been used before). I believe that there are citations to drifters used in fluvial and oceanographic studies but no detail or discussion is present of their capabilities or performance. This is odd given the space afforded to wireless in situ sensors within glaciology, much of which isn't really relevant to this study. I would recommend that the introduction of Lagrangian drifters is expanded and that the removal of any strictly unnecessary sections is considered.

2. Given that this paper introduces a new instrument, the methods section lacks a decent description of the drifter electronics or the sensor's physical construction. The drifter's sensors are described but there is no description of the microprocessor used or the method of data storage. No schematic is provided and the method of fabrication is not mentioned. Hence, many questions present themselves such as how is the microprocessor programmed and in what language? What is the sensor housing made from and how robust is it? Could it survive deployment in a subglacial channel? What water depth can the housing withstand? It would not be easy to replicate the experiment without further information and it is currently difficult to assess the limitations of the existing system

3. The description of the results is very verbose with many unnecessary explanations of standard statistical techniques and detailed descriptions of what is plotted in the figures. As such, the manuscript could be condensed with no loss of important detail. Please see specific comments below. Condensing the text may also highlight opportunities for minor restructuring (e.g. combining sections).

[Figure]

Minor comments

P2L8 – "has also been" rather than "was also"

P2L9 - delete "the" before "channel"

P2L21 – given the intention to discuss new methods, SF6 tracing should be mentioned (e.g. Chandler et al. 2013).

P2L25 - Andrews et al. (2014) also instrumented moulins and their results I would argue are more than encouraging. There are also a few other studies that are not cited so I suggest you use e.g. before the citations.

P2L29 - while this is arguably true, it could also be argued that the majority of data still comes from wired sensors. There have also been recent developments in wired sensors. I'm not sure this needs mentioning and I would recommend focusing the introduction on drifters rather than borehole sensors.

P2L34 – The sentence beginning "Drifters . . ." needs fragmenting, e.g. with commas. (Other sentences may benefit from this as well).

P3L16 - please state what you mean by "multimodal".

P3L29 – avoid the colloquial phrase 'already coming up'

F1 caption – change "pressure holes" to "holes for pressure transducers"

P7L3 - define POM

P7L7 - by total pressure do you mean what is normally referred to as gauge pressure, which is the pressure indicated by the gauge and not corrected for e.g. atmospheric pressure variability? What digital communications protocol do these sensors use? What resolution? Accuracy? More detail is required here.

P7L9 – 'linear calibration' rather than 'linearly rated'

P7L12 - please explain what is meant by a second order corrective algorithm. Is this

a second order polynomial? I realise this is described below but it could be clearer. If I follow right the zeroing is one-off so it's not right to say sub-diurnal variability is calibrated out as any post-zeroing variability in atmospheric pressure would not be corrected for.

P8L10 - more discussion of the BNO055 calibration would be worthwhile. My understanding is that this sensor self-calibrates continuously, which I expect has advantages and disadvantages with implications for the data collected. Is changing this sensor one of the future technical improvements you allude to below?

P5L2 - write out month in full

P12L8 – typos: extra "an" and on L13 an extra "in".

P12L21 - filtered how?

P13L14 – This section could be condensed by assuming the reader understands basic statistics and with the use of symbols and terminology. See below.

P13L24 – 'assess' should be 'assesses', though 'identifies' or similar may be a better word here. That said skewness and kurtosis should not need defining, as they are standard statistical techniques.

P13L25 – the terms 'magnetometer in the y-direction' and 'gyroscope in the y-direction' are somewhat awkward which makes it difficult to read. Perhaps use symbols instead? E.g. My, Gy. Euler angles are often referred to as yaw, pitch and roll and have standard symbols.

P13L26 – "are slightly skewed towards values above the mean" can be written in less words as "are positively skewed".

P13L30 – high kurtosis is referred to as 'leptokurtic'. A kurtosis which is nearly Gaussian can be referred to as no kurtosis (or almost no kurtosis). This section can be condensed significantly if these terms are used.

P15L3 – delete 'data set' as its not necessary. The manuscript would be easier to read if unnecessary words were removed.

P16L4 – you don't need to explain Pearson's correlation coefficient. Scientific papers would become impractically long if every standard technique was introduced. If a non-standard technique is used by all means describe it in the methods (not the results). It's also not necessary to list the classifications of Cohen et al. (1992) in full. Just say that you use their classifications in a single sentence and give the citation. If the reader is interested they can look it up. I would also recommend avoiding the style of describing what the figures show, as you do on L9-10. Instead I would recommend the style of making a statement or argument followed by the figure reference. This paragraph could be condensed to a few sentences without any loss of important detail. As it stands there are seven sentences before a result is described.

P18L5 and P23L3 and other occurrences – Phrases such as "the next plot in Figure 8" and "as shown in Figure 9" can be shortened by just giving the figure reference in brackets.

P23L2 and other occurrences – the first sentence here is methods and should not need repeating here.

P25L5 – Referring to sample sizes on P14L9 you state that "These high numbers are however not necessarily an indicator of sensor accuracy, but rather an indicator of spatial and temporal flow variability", which obviously casts doubt on whether the calculations of a required sample size are useful at all. However, here you refer to the required sample size calculations again to conclude that such experiments will require "a significant number of deployments". Which of these is your preferred interpretation of your analysis on sample sizes?

P25L8 – Do you mean ($p > 0.05$) rather than less than?

P25L8/9 – how will technical improvements to the drifter reduce the number of deployments required? Please be specific. What are the specific issues with the drifter presented here? What needs to be improved?

References

Andrews, L. C., Catania, G. A., Hoffman, M. J., Gulley, J. D., Luthi, M. P., Ryser, C., Hawley, R. L. & Neumann, T. A. 2014. Direct observations of evolving subglacial drainage beneath the Greenland Ice Sheet, Nature, 514, 80-83

Chandler, D.,Wadham, J., Lis, G., Cowton, T., Sole, A., Bartholomew, I., Telling, J., Nienow, P., Bagshaw, E., Mair, D., Vinen, S. & Hubbard, A. 2013. Evolution of the subglacial drainage system beneath the Greenland Ice Sheet revealed by tracers, Nature Geoscience, 6, 195-198

---

## Author Comment (AC1) · 2 Oct 2019

We would like to thank Elizabeth Bagshaw for reviewing our manuscript and providing helpful and constructive feedback, which will help to improve the current manuscript.

In the following we respond to the comments and outline how we will address these in the revision of the manuscript.

Referee comments are presented in **bold and italic**, and our replies follow directly thereafter.

**General comments**

*This paper gives a comprehensive account of a challenging experiment to assess the repeatability and usability of data collected by a drifting sensor. It should be noted that experiments including field testing of new engineering techniques in extreme environments are exceptionally challenging, and the authors should be congratulated on their successful deployment. The paper is well-written, and gives a comprehensive review of much of the supporting literature.*

Thank you for this positive overall judgment.

*However, the scope of the paper is somewhat confusing to the reader, since the problem is framed as a subglacial experiment, yet the data are confined to the supraglacial environment. This in no way invalidates the results, but I believe that the paper would be more appealing to the target audience if the supraglacial scope were made clear from the first paragraph and in the abstract. The subglacial deployment may be the ultimate goal of the project, but the current state of the science and engineering is limited to the supraglacial. I advise that the introductory paragraphs and abstract are focussed on supraglacial literature, with some additional references as necessary (for example, Decaux et al 2019 cited later). The subglacial material can then be moved to later in the discussion, to make clear how the supraglacial results can be utilised and developed in the future.*

We agree. The scope of the current iteration of this manuscript was obviously not made clear enough, as also noted by referee 2. The subglacial deployment is indeed the goal of this project, but not part of this manuscript. We believe therefore that the focus should not be supraglacial alone but on the overall range of potential applications

of the drifter technology. We will shorten and rephrase the introduction and the abstract accordingly to make the scope of the manuscript clearer and avoid confusion for the reader.

In order to understand the data, which can be acquired with the proposed drifter technology, it was crucial for the study to address practical goals associated with determining the suitability of the method for wider implementation:

The first goal was to conduct a comprehensive statistical assessment of repeated field deployments to determine the limits of the proposed methods, and to determine the expected measurement performance of the devices for future studies. The second was to investigate the multi-modal drifter time series and it's relation to the physical conditions experienced along the flow path. Finally, the third goal was to determine possible linkages between flow path geometry and time series signals to identify distinct properties of channel geometry. This can either be achieved by using supraglacial channels or englacial/subglacial channels whose length and geometry are known apriori.

*I like the use of statistics to validate the sensor performance, and the realism in relating the statistical results to logistical practicality. However, the actual purpose of the paper is not entirely clear in this iteration – is it an engineering test, a sensor validation exercise, or does it reveal a previously unknown glaciological phenomenon? All of these are valid outcomes, but the introduction and particularly the abstract should be better focused to demonstrate that experimental purpose to the reader. The paper could also be shortened by moving some of the tables to supplementary info (see below).*

The current paper does not show any previously unknown glaciological phenomena. The focus is to demonstrate the general feasibility, provide a statistical performance assessment, and to validate the data produced by the sensors. We concur with the

reviewer's comment, and will reformulate the introduction and the abstract to make this clear. We will also shorten the manuscript by moving some of the figures and tables to the supplementary material, following the reviewer's suggestions.

**Specific comments**

*Figure 1: please could you include a labelled photo of the sensors? It would be great to see them in a bit more detail.*

We thank the reviewer for this suggestion. We will add a labeled photo of the sensors.

*Figure 2: Place names are illegible on the map, and the features of the glacier need some labels in c).*

The notice of this error is appreciated. We will improve the figure and add additional labels in c).

*It would be useful to know where the net was situated, for example, and perhaps have some accompanying photos of the deployment/recovery sites.*

We will add this information in the manuscript/ on the figure.

*Could you comment on the feasibility of the net recovery in a larger system, and if the sensors are destined for the subglacial system, on the robustness of the net methods with debris and bedload transport in the flow?*

This is an excellent question. Our current net method seems to us to be at the moment only feasible for smaller systems. The main problem with the net method is the clogging of the net with ice on supraglacial channels. Similar problems occur in

front of subglacial systems where the additional bedload transport also clogs the net. High discharge and flow velocity make the installation and handling of the nets difficult as well. We will add a comment about this in the manuscript.

**Table 1: Could transmission distance (if relevant) be a separate column rather than a comment?**

Transmission distance is only mentioned once in the comment for the Smeets et al. (2012) reference. Making a separate column, would contain a majority of empty entries for the other references. So we would prefer to leave it as it is.

**P3, L23: Is 500 Euro truly low cost? This is a subjective term.**

Agreed. This is a subjective term and we will remove this.

**P10, L20: Please include an estimate of the range of discharge variability.**

We did not measure it, so any estimation would be wrong. We will add an according comment.

**Figure 3 doesn't really add much to the paper, it could be removed to save space without detriment, since the workflow is not unusual and is described in the text.**

Agreed, we will remove this figure.

**P12, L25: please define the 'features of subglacial channels'**

A more detailed definition will be provided. In our context, features of subglacial channels include step-riser, glide and chute sequences, and the associated signals in the multi-modal time series data (e.g. sudden change in linear acceleration)

which correspond to these morphological sequences. In addition, features can also correspond to changes in the local pressure field, indicating regions where the total water pressure increases or decreases along the flow path.

**Figure 4 and Tables 3 and 5 may be better placed in supplementary info, since their content is only of interest to a very specific audience and the paper is rather long.**

We will move them to the supplementary info.

**Table 4 and P14, L7: what is the 'required size' for a subglacial deployment? Unclear how these measurements are extrapolated to the subglacial system: just because the drifter can move through an open supraglacial stream doesn't necessarily infer that it will pass through the subglacial environment.**

This number was inferred from the deployment of drifter dummies with the same size and buoyancy than the real drifters through englacial systems. We will clarify this in more detail.

**Figure 10: Can other sensor data be added to this figure? It would be very useful to see the accelerometer data plotted alongside. The IMU accelerometer method is really exciting, so if the data could be demonstrated alongside the pressure and photographs, it would really contribute something valuable to the field.**

Thanks for this suggestion. We will add additional IMU data to the figure.

**P25, L11-25: What were you hoping to determine with this dataset? It seems that you have proven that the technology and the sensor set work (which is great!), so can you relate this to the flowpaths? How do the data relate to visual ob-**

*servations? If you hope to use this to visualise subglacial systems, then it is important to relate the sensor data (of which you have a considerable quantity) from the supraglacial system to visual observations where you can. Then you can demonstrate how this might be used in the subglacial environment. 'We need more data' seems a bit of a cop-out! What precision do you need to obtain scientifically useful data?*

We were hoping to show the repeatability of the dataset. This step is an important first step (and the focus of this study) towards retrieving information about subglacial drainage from our drifters. We were also attempting to do what you are pointing out here, but having only thirteen data points and no channel geometries (due to temporal and man-power constraints) seems not very convincing to us to make conclusive statements about the signal features of certain channel geometries. We will however elaborate more on this topic to improve clarity.

*P26, L26: What is 'satisfactory performance'? This is very subjective. What did this experiment hope to achieve, and did you do it? Was it field testing of the casing, the transport method, or of the sensor performance, or of the usefulness of the data to characterise the supraglacial flowpath? Or of future subglacial deployment? Please be specific – this is an excellent engineering test, but subjectivity in appraisal should be avoided.*

Thanks for pointing this out. We will remove any subjectivity and clarify the scope of this experiment.

*Table 6 isn't terribly useful.*

We will remove it.

---

## Author Comment (AC2) · 2 Oct 2019

First of all we would like to thank Samuel Doyle for reviewing our manuscript and provide helpful and constructive feedback, which will help to improve the paper. In the following we present our responses to the referee comments and how we will address these in the revision of the manuscript.

The referee comments are presented in ***bold and italic***, our replies follow immediately thereafter.

[Figure]

**Overall comments:**

*Alexander et al. present a statistical assessment of the performance of a new sensing system – a Lagrangian drifter – for glacier hydrological experiments. They report results from repeated tests in a supraglacial channel and suggest (though never that directly) that there may be future potential for deploying the drifter within the subglacial environment. The sensor system is novel and this manuscript makes an important contribution to the very limited literature on Lagrangian drifters in glaciology.*

We thank the reviewer for his positive judgment.

*Although it is verbose, the paper is generally well-written.*

We agree, and will shorten the manuscript in the following revision, please see also the response to RC 1.

*The figures and tables are clear, though the number of tables and figures within the manuscript could be reduced.*

The total number of figures and tables will be reduced by either removing them, or adding them to the supplement. See also the response to RC 1.

*Citations are appropriate, however, it is unclear why the introduction focuses (e.g. Table 1) on wireless in situ sensor systems, which are not really relevant, while previously published drifter studies from fluvial and oceanographic studies are not discussed in detail.*

The idea was to give a general overview of different available technologies to measure in the **subglacial** environments and then present drifters as additional possibility. We

will add additional clarification in the introduction.

**Major comments:**

*1. With the exception of the "Moulin Explorer" and the eTracer, the introduction lacks a section describing what drifters are currently available (or have been used before). I believe that there are citations to drifters used in fluvial and oceanographic studies but no detail or discussion is present of their capabilities or performance. This is odd given the space afforded to wireless in situ sensors within glaciology, much of which isn't really relevant to this study. I would recommend that the introduction of Lagrangian drifters is expanded and that the removal of any strictly unnecessary sections is considered.*

There are indeed a wealth of different drifter platforms available for fluvial and oceanographic studies. However, there are very few which are specially designed to withstand harsh physical environments, especially the conditions faced during subglacial deployments. Our focus is on drifters for glaciological studies, and a complete review including river and oceanographic drifter platforms would be a stand-alone contribution. Too maintain the focus and scope of this manuscript, we will shorten the current introduction to make it more clear and concise, and at the same time include additional references to general drifter capabilities in other fields.

*2. Given that this paper introduces a new instrument, the methods section lacks a decent description of the drifter electronics or the sensor's physical construction. The drifter's sensors are described but there is no description of the microprocessor used or the method of data storage. No schematic is provided and the method of fabrication is not mentioned.*

We will add additional information about the fabrication process, as well as the used
microprocessor and the method of data storage. We will additionally also modify the current drifter illustration (see also comment to RC1).

*Hence, many questions present themselves such as how is the microprocessor programmed and in what language?*

We will add this information.

*What is the sensor housing made from and how robust is it? Could it survive deployment in a subglacial channel?*

The sensor housing is made of a polycarbonate tube and the endcaps are made from Polyoxymethylene plastic, and can withstand 3000 g of impact. They were originally designed and have been successfully deployed to measure conditions in large-scale hydropower turbines. Successful survival in a subglacial channel was meanwhile proven during new field tests in 2019. The latter tests will be incorporated as part of a future subglacial study using newer, smaller versions of the drifters . However, we will add a comment to our present manuscript to confirm robustness of the drifters from our new field tests.

*What water depth can the housing withstand? It would not be easy to replicate the experiment without further information and it is currently difficult to assess the limitations of the existing system.*

The pressure sensors are tested in a laboratory barochamber up to 55 m water depth, and the measurement limitation results from the measurement range of the pressure sensor, rather than the housing. We will give a more detailed description of the platform as stated in the answers above.
*3. The description of the results is very verbose with many unnecessary explanations of standard statistical techniques and detailed descriptions of what is plotted in the figures. As such, the manuscript could be condensed with no loss of important detail. Please see specific comments below. Condensing the text may also highlight opportunities for minor restructuring (e.g. combining sections).*

We will cut down the unnecessary explanations and condense the result section considerably. Additionally several of the figures and the tables will be removed from the main manuscript or moved to the supplement.

**Minor comments:**

*P2L8 – "has also been" rather than "was also"*

Will be changed.

*P2L9 - delete "the" before "channel"*

Will be changed.

*P2L21 – given the intention to discuss new methods, SF6 tracing should be mentioned (e.g. Chandler et al. 2013).*

Thanks. We will include it.

*P2L25 - Andrews et al. (2014) also instrumented moulins and their results I would argue are more than encouraging. There are also a few other studies that are not cited so I suggest you use e.g. before the citations.*

We are thankful to the reviewer for making us aware of this study. We will include it in our references and also include "e.g." as suggested.

*P2L29 - while this is arguably true, it could also be argued that the majority of data still comes from wired sensors. There have also been recent developments in wired sensors. I'm not sure this needs mentioning and I would recommend focusing the introduction on drifters rather than borehole sensors.*

We will still mention the borehole sensors in the introduction but condense it considerably and shift the focus towards drifters.

*P2L34 – The sentence beginning "Drifters : : :" needs fragmenting, e.g. with commas. (Other sentences may benefit from this as well).*

Will be done.

*P3L16 - please state what you mean by "multimodal".*

Will be done.

*P3L29 – avoid the colloquial phrase 'already coming up'*

We will remove it.

*F1 caption – change "pressure holes" to "holes for pressure transducers"*

Will be done.

*P7L3 - define POM*

Will be done.

**P7L7 - by total pressure do you mean what is normally referred to as gauge pressure, which is the pressure indicated by the gauge and not corrected for e.g. atmospheric pressure variability? What digital communications protocol do these sensors use? What resolution? Accuracy? More detail is required here.**

We will add more detail.

**P7L9 – 'linear calibration' rather than 'linearly rated'**

We will change this.

**P7L12 - please explain what is meant by a second order corrective algorithm. Is this a second order polynomial? I realise this is described below but it could be clearer. If I follow right the zeroing is one-off so it's not right to say sub-diurnal variability is calibrated out as any post-zeroing variability in atmospheric pressure would not be corrected for.**

We will clarify this.

**P8L10 - more discussion of the BNO055 calibration would be worthwhile. My understanding is that this sensor self-calibrates continuously, which I expect has advantages and disadvantages with implications for the data collected. Is changing this sensor one of the future technical improvements you allude to below?**

This is a very good question. There are major trade-offs between using the BNO055 and other IMUs which do not have on-chip sensor fusion. The major benefit of the BNO055 is indeed that the absolute orientation is calculated in real-time. The major

downside is that the calibration and sensor fusion used in this procedure are "black box" in the sense that Bosch has not released the algorithms used. The next generation of smaller, less expensive drifters will incorporate newer IMUs which have lower energy consumption, as well as allow for absolute orientation in post-processing.

In addition, we will provide references to the accuracy of the BNO055 sensors, which are reported in several peer-reviewed journals, (mostly related to human kinematic studies).

*P5L2 - write out month in full*

Will be done.

*P12L8 – typos: extra "an" and on L13 an extra "in".*

Thanks for pointing this out. Will be changed.

*P12L21 - filtered how?*

We will add the additional information on the filtering.

*P13L14 – This section could be condensed by assuming the reader understands basic statistics and with the use of symbols and terminology. See below.*

We will shorten this section.

*P13L24 – 'assess' should be 'assesses', though 'identifies' or similar may be a better word here. That said skewness and kurtosis should not need defining, as they are standard statistical techniques.*

We will change the wording and remove the definitions.

*P13L25 – the terms 'magnetometer in the y-direction' and 'gyroscope in the y-direction' are somewhat awkward which makes it difficult to read. Perhaps use symbols instead? E.g. My, Gy. Euler angles are often referred to as yaw, pitch and roll and have standard symbols.*

Agreed. We will change it accordingly.

*P13L26 – "are slightly skewed towards values above the mean" can be written in less words as "are positively skewed".*

We will use this short form.

*P13L30 – high kurtosis is referred to as 'leptokurtic'. A kurtosis which is nearly Gaussian can be referred to as no kurtosis (or almost no kurtosis). This section can be condensed significantly if these terms are used.*

We will use those terms instead to shorten this section. Thanks.

*P15L3 – delete 'data set' as its not necessary. The manuscript would be easier to read if unnecessary words were removed.*

We will go through the manuscript carefully and remove this and other unnecessary words.

*P16L4 – you don't need to explain Pearson's correlation coefficient. Scientific papers would become impractically long if every standard technique was introduced. If a nonstandard technique is used by all means describe it in the meth-*

*ods (not the results). It's also not necessary to list the classifications of Cohen et al. (1992) in full. Just say that you use their classifications in a single sentence and give the citation. If the reader is interested they can look it up. I would also recommend avoiding the style of describing what the figures show, as you do on L9-10. Instead I would recommend the style of making a statement or argument followed by the figure reference. This paragraph could be condensed to a few sentences without any loss of important detail. As it stands there are seven sentences before a result is described.*

We will shorten this and other paragraphs accordingly. Thanks.

*P18L5 and P23L3 and other occurrences – Phrases such as "the next plot in Figure 8" and "as shown in Figure 9" can be shortened by just giving the figure reference in brackets.*

Will be done.

*P23L2 and other occurrences – the first sentence here is methods and should not need repeating here.*

We will remove it.

*P25L5 – Referring to sample sizes on P14L9 you state that "These high numbers are however not necessarily an indicator of sensor accuracy, but rather an indicator of spatial and temporal flow variability", which obviously casts doubt on whether the calculations of a required sample size are useful at all. However, here you refer to the required sample size calculations again to conclude that such experiments will require "a significant number of deployments". Which of these is your preferred interpretation of your analysis on sample sizes?*

We will clarify the interpretation here as well.

*P25L8 – Do you mean (p > 0.05) rather than less than?*

Yes, it should indeed be p>0.05. Thanks for catching this mistake.

*P25L8/9 – how will technical improvements to the drifter reduce the number of deployments required? Please be specific. What are the specific issues with the drifter presented here? What needs to be improved?*

Technical improvements of the drifters will reduce the signal to noise ratio of the drifters. In addition, the number of deployments can be decreased with improved field deployment and recovery methods. We will briefly address these additional topics in the improved version of our manuscript.

---

## Author Response (AR1)

**Response to referee comments**

**Multi-modal sensing drifters as a tool for repeatable glacial hydrology flow path measurements**

Andreas Alexander[1,2], Maarja Kruusmaa[3,5], Jeffrey A. Tuhtan[3], Andrew J. Hodson[2,4], Thomas V. Schuler[1,2], Andreas Kääb[1]

[1]Department of Geosciences, University of Oslo, 0316 Oslo, Norway
[2]Department of Arctic Geology, The University Centre in Svalbard, 9171 Longyearbyen, Norway
[3]Centre for Biorobotics, Tallinn University of Technology, 12618 Tallinn, Estonia
[4]Department of Environmental Sciences, Western Norway University of Applied Sciences, 6856 Sogndal, Norway
[5]Centre for Autonomous Marine Operations and Systems, Norwegian University of Science and Technology, 7491 Trondheim, Norway

*Correspondence to*: Andreas Alexander (andreas.alexander@geo.uio.no)

We would like to thank the reviewers for their constructive feedback and valuable input that certainly helped to improve this manuscript. The manuscript has been shortened considerable. All figures have been revised to improve readability. The detailed responses to the reviewer comments are presented below. A mark-up version of the manuscript, showing the changes made in response to the referee's comments follows thereafter.

**Response to RC 1**

We would like to thank Elizabeth Bagshaw for reviewing our manuscript and providing helpful and constructive feedback, which helped to improve our manuscript. In the following we respond to the comments and outline how we addressed these in the revision of the manuscript. Referee comments are presented in *italic*, and our replies follow directly thereafter.

**General changes**

All figures are revised to improve readability, and annotations have been added to label the step-pool examples. The introduction has been substantially revised and includes new references pertaining to supraglacial systems as well as for oceanographic / river drifters.

**General comments**

*This paper gives a comprehensive account of a challenging experiment to assess the repeatability and usability of data collected by a drifting sensor. It should be noted that experiments including field testing of new engineering techniques in extreme environments are exceptionally challenging, and the authors should be congratulated on their successful deployment. The paper is well-written, and gives a comprehensive review of much of the supporting literature.*

Thank you for this positive overall judgment.

*However, the scope of the paper is somewhat confusing to the reader, since the problem is framed as a subglacial experiment, yet the data are confined to the supraglacial environment. This in no way invalidates the results, but I believe that the paper would be more appealing to the target audience if the supraglacial scope were made clear from the first paragraph and in the abstract. The subglacial deployment may be the ultimate goal of the project, but the current state of the science and engineering is limited to the supraglacial. I advise that the introductory paragraphs and abstract are focussed on supraglacial literature, with some additional references as necessary (for example, Decaux et al 2019 cited later). The subglacial material can then be moved to later in the discussion, to make clear how the supraglacial results can be utilised and developed in the future.*

We reworked the abstract and the introduction to make the scope of the study more clear. We additionally added further references related to supraglacial systems, and reduced the focus on subglacial investigations.

*I like the use of statistics to validate the sensor performance, and the realism in relating the statistical results to logistical practicality. However, the actual purpose of the paper is not entirely clear in this iteration – is it an engineering test, a sensor validation exercise, or does it reveal a previously unknown glaciological phenomenon? All of these are valid outcomes, but the introduction and particularly the abstract should be better focused to demonstrate that experimental purpose to the reader. The paper could also be shortened by moving some of the tables to supplementary info (see below).*

We shortened the manuscript and clarified the scope of the work and research objectives.

**Specific comments**

*Figure 1: please could you include a labelled photo of the sensors? It would be great to see them in a bit more detail*

We added an additional figure (figure 2), illustrating the sensor components in greater detail.

*Figure 2: Place names are illegible on the map, and the features of the glacier need some labels in c).*

We changed the map in a) and added additional labels and pictures in c).

*It would be useful to know where the net was situated, for example, and perhaps have some accompanying photos of the deployment/recovery sites.*

Added to the figure in c).

*Could you comment on the feasibility of the net recovery in a larger system, and if the sensors are destined for the subglacial system, on the robustness of the net methods with debris and bedload transport in the flow?*

We added a comment about the net recovery in the Discussion part of our improved version of the manuscript (P19 L17-22).

*Table 1: Could transmission distance (if relevant) be a separate column rather than a comment?*

Transmission distance is only mentioned once in the comment for the Smeets et al. (2012) reference. Making a separate column would therefore leave the majority of entries empty for the other references. So we would prefer to leave it as it is. We however moved this table to the supplementary material.

*P3, L23: Is 500 Euro truly low cost? This is a subjective term.*

We removed it.

*P10, L20: Please include an estimate of the range of discharge variability*

We did not measure it, so any estimation would be speculative. We added an according comment (P9, L6).

*Figure 3 doesn't really add much to the paper, it could be removed to save space without detriment, since the workflow is not unusual and is described in the text*

Removed.

*P12, L25: please define the 'features of subglacial channels'*

We added additional information (P10, L29-30).

*Figure 4 and Tables 3 and 5 may be better placed in supplementary info, since their content is only of interest to a very specific audience and the paper is rather long*

Moved to the supplementary information.

*Table 4 and P14, L7: what is the 'required size' for a subglacial deployment? Unclear how these measurements are extrapolated to the subglacial system: just because the drifter can move through an open supraglacial stream doesn't necessarily infer that it will pass through the subglacial environment*

We added additional information to clarify (P11, L15-16).

*Figure 10: Can other sensor data be added to this figure? It would be very useful to see the accelerometer data plotted alongside. The IMU accelerometer method is really exciting, so if the data could be demonstrated*

*alongside the pressure and photographs, it would really contribute something valuable to the field.*

We added the accelerometer data to this figure (P18, F9). Additionally we also marked the step-pool sequence on the other plots (P15, F6; P16, F7; P17, F8).

*P25, L11-25: What were you hoping to determine with this dataset? It seems that you have proven that the technology and the sensor set work (which is great!), so can you relate this to the flowpaths? How do the data relate to visual observations? If you hope to use this to visualize subglacial systems, then it is important to relate the sensor data (of which you have a considerable quantity) from the supraglacial system to visual observations*

10 *where you can. Then you can demonstrate how this might be used in the subglacial environment. 'We need more data' seems a bit of a cop-out! What precision do you need to obtain scientifically useful data?*

We added additional information to clarify this section (P19, L26-29).

15 *P26, L26: What is 'satisfactory performance'? This is very subjective. What did this experiment hope to achieve, and did you do it? Was it field testing of the casing, the transport method, or of the sensor performance, or of the usefulness of the data to characterise the supraglacial flowpath? Or of future subglacial deployment? Please be specific – this is an excellent engineering test, but subjectivity in appraisal should be avoided.*

20 Thank you for this comment, we went through the manuscript and removed subjective statements, and clarified the scope of the experiment.

*Table 6 isn't terribly useful.*

25 Agreed, the table has been removed.

**Response to RC 2**

We would like to thank Samuel Doyle for reviewing our manuscript and provide helpful and constructive feedback, which will helped to improve the manuscript. In the following we present our responses to the referee comments and how we addressed these in the revision of the manuscript. The referee comments are presented in
5    bold and *italic*, our replies follow immediately thereafter.

**General changes**

All figures are revised to improve readability, and annotations have been added to label the step-pool example. The introduction has been substantially revised and includes new references pertaining to supraglacial systems as
10   well as for oceanographic / river drifters.

**Overall comments**

*Alexander et al. present a statistical assessment of the performance of a new sensing system – a Lagrangian drifter – for glacier hydrological experiments. They report results from repeated tests in a supraglacial channel and suggest (though never that directly) that there may be future potential for deploying the drifter within the*
15   *subglacial environment. The sensor system is novel and this manuscript makes an important contribution to the very limited literature on Lagrangian drifters in glaciology.*

We thank the reviewer for his positive judgment.

20   *Although it is verbose, the paper is generally well-written.*

We shortened the manuscript. See also the answer to major comment 3.

*The figures and tables are clear, though the number of tables and figures within the manuscript could be*
25   *reduced.*

Removed: Figure 3, Table 6.
Supplement: Table 2, Figure 4, Table 3, Table 5

30   *Citations are appropriate, however, it is unclear why the introduction focuses (e.g. Table 1) on wireless in situ*

*sensor systems, which are not really relevant, while previously published drifter studies from fluvial and oceanographic studies are not discussed in detail.*

We understand the comment, and have revised the introduction accordingly. The idea was to give a general overview of different available technologies to measure in the subglacial environments and then present drifters as additional possibility. We added additional references for fluvial and oceanographic drifter applications. And further clarified the introduction.

**Major comments**

*1. With the exception of the "Moulin Explorer" and the eTracer, the introduction lacks a section describing what drifters are currently available (or have been used before). I believe that there are citations to drifters used in fluvial and oceanographic studies but no detail or discussion is present of their capabilities or performance. This is odd given the space afforded to wireless in situ sensors within glaciology, much of which isn't really relevant to this study. I would recommend that the introduction of Lagrangian drifters is expanded and that the removal of any strictly unnecessary sections is considered.*

We removed unnecessary parts from the introduction and added information about fluvial and oceanographic drifters and their capabilities.

*2. Given that this paper introduces a new instrument, the methods section lacks a decent description of the drifter electronics or the sensor's physical construction. The drifter's sensors are described but there is no description of the microprocessor used or the method of data storage. No schematic is provided and the method of fabrication is not mentioned.*

The devices are made from more than 100 different electronic and mechanical components, a description of the complex fabrication process is therefore out of scope for this manuscript. A detailed technical overview of the drifter electronics is being prepared for the IEEE Sensors journal. We feel that a detailed overview of the electronics and communications, including schematics requires a stand-alone publication.

*Hence, many questions present themselves such as how is the microprocessor programmed and in what language?*

Information has been added as requested.

*What is the sensor housing made from and how robust is it? Could it survive deployment in a subglacial channel?*

We added additional information.

*What water depth can the housing withstand? It would not be easy to replicate the experiment without further information and it is currently difficult to assess the limitations of the existing system.*

These are practical questions which we overlooked, and appreciate the comment. Additional details have been added.

*3. The description of the results is very verbose with many unnecessary explanations of standard statistical techniques and detailed descriptions of what is plotted in the figures. As such, the manuscript could be condensed with no loss of important detail. Please see specific comments below. Condensing the text may also highlight opportunities for minor restructuring (e.g. combining sections).*

We cut down unnecessary explanations and condensed the result section. Several figures and tables have been removed or moved to the supplementary material.
The text has been condensed in the following:
Section 3.2, 3.3 and 3.4: Merged and condensed from 6 to 2 paragraphs.
Section 3.5 merged with 3.6: Condensed from 7 paragraphs to 3.
Section 3.7: Condensed from 2 paragraphs to 1.

**Minor comments:**

*P2L8 – "has also been" rather than "was also"*

Amended as suggested.

*P2L9 - delete "the" before "channel"*

Addressed.

*P2L21 – given the intention to discuss new methods, SF6 tracing should be mentioned (e.g. Chandler et al. 2013). P2L25 - Andrews et al. (2014) also instrumented moulins and their results I would argue are more than encouraging. There are also a few other studies that are not cited so I suggest you use e.g. before the citations.*

We have revised the references to include additional studies to provide a wider range of background material, with less focus on the subglacial environment.

*P2L29 - while this is arguably true, it could also be argued that the majority of data still comes from wired sensors. There have also been recent developments in wired sensors. I'm not sure this needs mentioning and I would recommend focusing the introduction on drifters rather than borehole sensors.*

We removed the part of the introduction in question and limited the discussion of borehole sensors to Table 1 of the supplementary material, as suggested we have focused specifically on drifters.

*P2L34 – The sentence beginning "Drifters : : :" needs fragmenting, e.g. with commas. (Other sentences may benefit from this as well).*

Changed following the reviewer's suggestion.

*P3L16 - please state what you mean by "multimodal".*

Stated as requested (P3, L11).

*P3L29 – avoid the colloquial phrase 'already coming up'*

Removed as suggested.

*F1 caption – change "pressure holes" to "holes for pressure transducers"*

Amended accordingly.

*P7L3 - define POM*

POM is now referred using the full name, Polyoxymethylene.

*P7L7 - by total pressure do you mean what is normally referred to as gauge pressure, which is the pressure indicated by the gauge and not corrected for e.g. atmospheric pressure variability? What digital communications protocol do these sensors use? What resolution? Accuracy? More detail is required here.*

We have added additional detail related to the total pressure sensor as requested (P4, L7-18).

*P7L9 – 'linear calibration' rather than 'linearly rated'*

Removed as suggested.

*P7L12 - please explain what is meant by a second order corrective algorithm. Is this a second order polynomial? I realise this is described below but it could be clearer. If I follow right the zeroing is one-off so it's not right to say sub-diurnal variability I calibrated out as any post-zeroing variability in atmospheric pressure would not be corrected for.*

Second-order refers to a two-stage process, and has been amended in the text accordingly by replacing "second order" with "two-stage". The algorithm first takes into account device-specific correction coefficients, specified by the manufacturer. In a second step, the device's temperature is used to output the corrected total pressure reading depending on the temperature range. The algorithm is provided by the manufacturer in their datasheet, page 7: https://www.te.com/global-en/product-CAT-BLPS0059.html

We added this information to the manuscript.

*P8L10 - more discussion of the BNO055 calibration would be worthwhile. My understanding is that this sensor self-calibrates continuously, which I expect has advantages and disadvantages with implications for the data collected. Is changing this sensor one of the future technical improvements you allude to below?*

We added additional information and provided references to the accuracy of the Bosch fusion algorithm.

*P5L2 - write out month in full*

Corrected as suggested.

*P12L8 – typos: extra "an" and on L13 an extra "in".*

Amended.

*P12L21 - filtered how?*

Additional information added (P10, L24-25).

*P13L14 – This section could be condensed by assuming the reader understands basic statistics and with the use of symbols and terminology. See below.*

The section has been shortened from three paragraphs down to one. Figure 4 and Table 3 were also removed and the section was merged with section 3.3. and 3.4. Section 3.2, section 3.3 and section 3.4 had 6 paragraphs in total and are now merged into two paragraphs.

*P13L24 – 'assess' should be 'assesses', though 'identifies' or similar may be a better word here. That said skewness and kurtosis should not need defining, as they are standard statistical techniques.*

We removed this from the text.

*P13L25 – the terms 'magnetometer in the y-direction' and 'gyroscope in the y-direction' are somewhat awkward which makes it difficult to read. Perhaps use symbols instead? E.g. My, Gy. Euler angles are often referred to as yaw, pitch and roll and have standard symbols.*

The terms have been amended accordingly and also included in figure 1.

*P13L26 – "are slightly skewed towards values above the mean" can be written in less words as "are positively skewed".*

The text has been removed to improve clarity.

*P13L30 – high kurtosis is referred to as 'leptokurtic'. A kurtosis which is nearly Gaussian can be referred to as no kurtosis (or almost no kurtosis). This section can be condensed significantly if these terms are used.*

This section has been removed.

*P15L3 – delete 'data set' as its not necessary. The manuscript would be easier to read if unnecessary words were removed.*

We removed unnecessary words from the manuscript.

*P16L4 – you don't need to explain Pearson's correlation coefficient. Scientific papers would become impractically long if every standard technique was introduced. If a nonstandard technique is used by all means describe it in the methods (not the results). It's also not necessary to list the classifications of Cohen et al. (1992) in full. Just say that you use their classifications in a single sentence and give the citation. If the reader is interested they can look it up. I would also recommend avoiding the style of describing what the figures show, as you do on L9-10. Instead I would recommend the style of making a statement or argument followed by the figure reference. This paragraph could be condensed to a few sentences without any loss of important detail. As it stands there are seven sentences before a result is described.*

We shortened this and other paragraphs in the results section accordingly.

*P18L5 and P23L3 and other occurrences – Phrases such as "the next plot in Figure 8" and "as shown in Figure 9" can be shortened by just giving the figure reference in brackets.*

We amended the wording to shorten the text as suggested.

*P23L2 and other occurrences – the first sentence here is methods and should not need repeating here.*

We removed all sections, which described methods.

*P25L5 – Referring to sample sizes on P14L9 you state that "These high numbers are however not necessarily an indicator of sensor accuracy, but rather an indicator of spatial and temporal flow variability", which obviously casts doubt on whether the calculations of a required sample size are useful at all. However, here you refer to the required sample size calculations again to conclude that such experiments will require "a significant number of deployments". Which of these is your preferred interpretation of your analysis on sample sizes?*

We have clarified the interpretation on P14L9 as well as the discussion on P25L5.

*P25L8 – Do you mean ($p > 0.05$) rather than less than?*

Thank you for catching this error, it has been corrected.

*P25L8/9 – how will technical improvements to the drifter reduce the number of deployments required? Please be specific. What are the specific issues with the drifter presented here? What needs to be improved?*

Upgrades to the sensors to reduce the number of deployments are two-fold: First, the sensor electronics of the drifters will replace the BNO055 with a high-g accelerometer, as the current range is limited to +/- 16g and impact events can cause the accelerometer to become saturated. We believe the resulting data will improve the characterization of the signals, therefore reducing the number of deployments required. Second, the number of deployments can also be decreased with improved field deployment and recovery methods. Specifically, we are

planning on reducing the size of the sensors to < 5cm maximal dimension, to reduce the chances of them getting stuck on deployment. We have added additional information in the discussion section.

**Multi-modal sensing drifters as a tool for repeatable glacial hydrology flow path measurements**

[revised manuscript text omitted]

high sedimentation rates (Walder and Fowler, 1994) , abrasion (Haldorsen, 1981) and turbulent water flow (Kor et al., 1991) .
(e.g. Gleason et al., 2016) . Beginning in the early 2000s, new technologies have emerged (presented in more detail in Table ??),
which open new, promising pathways towards an improved understanding of subglacial hydrology (e.g. Martinez et al., 2004; Hart et al., 20
and are outlined in Table A1 of the supplementary material. Current methods for in-situ tests include Doppler current profiling

5   in supraglacial systems (e.g. Gleason et al., 2016) , dye tracing (e.g. Seaberg et al., 1988; Willis et al., 1990; Fountain, 1993;
Nienow et al., 1998; Hasnain et al., 2001; Schuler and Fischer, 2009), salt injection gauging (e.g. Willis et al., 2012), geophysi-
cal methods (e.g. Diez et al., 2019) , and direct observations are available from borehole instrumentation (e.g. Iken and Bindschadler, 1986;
There are also encouraging attempts to deploy sensors in moulins (Iken, 1972; Vieli et al., 2004) . Direct access of the glacier
base has been exploited to collect measurements at the subglacial laboratory in Engabreen, Norway (e.g. Cohen et al., 2006; Iverson et al., 2

10   well as at the Argentière glacier in the French Alps (e.g. Vivian and Bocquet, 1973; Goodman et al., 1979; Hantz and Lliboutry, 1983) .
and gas tracing (e.g. Chandler et al., 2013) .

    Most of the recent developments in sensors in glacial hydrology (see Table ?? for a detailed overview) have been focused on
devices which can perform borehole measurements which can be transferred wirelessly through the ice (e.g. Martinez et al., 2004; Hart et al

15   The major limitations of fixed position observations is that they have to be deployed via borehole, decreasing the chances to
enter a subglacial system, and that boring requires a high deployment cost. This has motivated the development of Lagrangian
sensors which move with the changing environment, thus providing a wider range of observational data at a substantially
lower deployment cost. Drifters are small Lagrangian drifters are small floating devices which passively follow the wa-
ter flow and are commonly used in large-scale surface flow studies and are able to to study flow in large rivers, lakes and

[revised manuscript text omitted]

Hart et al. (2006) Pressure, tilt angle, temperature, resistivity, strain gauge Borehole Wireless One year Yes Clast transport Installed in sediment, Glacsweb project

Behar et al. (2009) Pressure, temperature, 3D acceleration Moulin Iridium Short-term No Conference Abstract/ Sensor development Platform lost during deployment

5    Rose et al. (2009) Temperature, pressure, resistance, tilt Borehole Wireless, One year Yes Basal conditions Glacsweb project

Hart et al. (2011a) Temperature, water pressure, probe deformation, conductivity, tilt Borehole Wireless, One year Yes Till behaviour Glacsweb project

Hart et al. (2011b) Water pressure, probe deformation, conductivity, temperature Borehole Wireless, 1-2 years Yes Investigation

10   of glacier break-up Glacsweb project

Bagshaw et al. (2012) Radio beacon, pressure Moulin Wireless, Short-term Yes Drifter development Only feasibility test of drifters, stationary pressure recordings

Smeets et al. (2012) Water pressure Borehole Wireless, 10 years Yes Subglacial pressure Wireless transfer through up to ice thickness

Lishman et al. (2013) Acoustic attenuation Borehole Wireless, different frequencies Short-term Yes Acoustic communication through ice predicted to be most feasible frequency for communication

Bagshaw et al. (2014) Pressure Moulin Wireless, / Short-term Yes Development of drifters Stationary Cryoegg Bagshaw et al. (2014) Pre- temperature, conductivity Moulin Wireless, / Short-term Yes Development of driftersETracer drifter

5    Hart et al. (2015) Temperature, water pressure, probe Borehole Wireless, One year Yes Study subglacial/ englacial waterflow Glacsweb project

van de Wal et al. (2015) Pressure Borehole Wireless, One year Yes Ice velocities Platform from Smeets et al., 2012

How et al. (2017) Pressure, temperature, tilt Borehole Wireless, 7-14 months Yes Subglacial hydrology Platform from Smeets et al., 2012

10   Martinez et al. (2017) Passive seismics, 3D acceleration, digital compass, temperature Borehole Cabled/ Wireless, Short-term deployment Yes Glacier stick-slip Fixed location Geophones, Glacsweb project

Bagshaw et al. (2018) Pressure, conductivity, temperature Borehole Wireless, 3 months Yes Subsurface firn/ snow studies Merged Cryoegg and ETracer platform

Hart et al. (2019) Pressure, stress, conductivity, tilt, temperature Borehole Wireless, Up to 2 years Yes Glacier stick-slip

15   motion, till deformation Glacsweb project This study Pressure, acceleration, magnetic field, spinning rate, Euler angels Moulins, meltwater channels WiFi, after recovery Short-term Yes Drifter proof-of-concept and data repeatability assessment Reliability study, time series feature detection

[revised manuscript text omitted]

=9). This lead to the definition of a recovery and a utility rate for the drifter deployments:

$$\text{Recovery rate} = \frac{\text{Number of recovered dummies/ drifters}}{\text{Number of deployed dummies /drifters}} . \tag{2}$$

$$\text{Utility rate} = \text{Recovery rate} \cdot \text{Data usability rate} = \frac{\text{Number of recovered dummies/ drifters}}{\text{Number of deployed dummies/ drifters}} \cdot \frac{\text{Number of usable datasets}}{\text{Number of recovered drifters}} \tag{3}$$

$$\text{Utility rate} = \text{Recovery rate} \cdot \text{Data usability rate} \tag{4}$$

**Supraglacial** system from the second experiment with 5 drifters and a total of 55 deployments in 450 m long supraglacial channel:

Recovery rate $= \frac{55}{55} = 1.00$ .

Utility rate $= \frac{55}{55} \cdot \frac{40}{55} = 0.73$ .

**Subglacial/englacial** system, based on the assumption that the drifters can pass through the system, if the dummies pass through. The results from the first experiment with 5 dummies in the 2.5 km long channel is used:

Recovery rate $= \frac{4}{5} = 0.80$ .

Estimated total utility rate for **subglacial/englacial** deployments based on dummy deployment and data usability rate:
Utility rate $= \frac{4}{5} \cdot \frac{40}{55} = 0.58$ .

**3.2 Statistical evaluation**

25

 We calculated the ensemble statistics for all successful deployments
30 ~~calculations for Quaternions and Euler angles. Additionally, the normal distribution fitted to each dataset is plotted as a red line. Both the visual interpretation of the empirical probability distributions in Figure 4 as well as the mean values in Table ?? strongly indicate that the values for the acceleration, rate gyroscope and the quaternions are close to zero, meaning the drifters remained nearly motionless for long periods of time during measurement. This is due to many of the drifters being stuck in the channel for several minutes before they were dislodged and carried further by the current.~~

~~The skewness of the distributions indicates the asymmetry of the data around the origin. It therefore assess whether a dataset is symmetric or if it is deviating from a central tendency. The pressures, magnetometer in the y-direction, gyroscope in the x-direction and the Euler angles in both the x- and y-directions are slightly skewed towards values above the mean. The acceleration in the y-direction is more skewed towards positive values due to the orientation (facing down and into the direction~~
5

10

, the histograms and numerical values can be found in the supplementary material. The mean values and the standard deviations  were then used to estimate the required sample size to achieve a precision of the sample mean to be within ±10% of the  time-averages (i.e. $\varepsilon = 0.10$) for 95% of the time (i.e. $Z_{0.975} = 1.96$). The obtained sam-
15 ple size estimates were afterwards multiplied with the utility rates to estimate the required number of supraglacial and subglacial/englacial deployments. The mean pressure values were thereby corrected with the calculated air pressure of 941.8 hPa based on elevation (600 m) and air temperature on 07.08.2018. This calculation resulted in unrealistically

high required sample sizes (Table 2). These high numbers are however  composed of several components: One part is caused by the sensor accuracy and technical problems causing high variations in the measured data. The second part of the inaccuracy is due to spatial and temporal flow variability both between deployments, but also along the flow path. The lowest required sample size was for the pressure sensors and the magnetic field intensity magnitude. The latter should however also be corrected by the value of the local magnetic field strength and the number of required deployments is therefore likely to be higher.

Distance and similarity measures were used to test the repeatability of the datasets. All calculated values for every sensor modality and statistical measure (Chi Squared Error, Kullback Leiber divergence, mean average error, mean squared error and data ranged normalized root mean square) are close to zero, thus indicating a high repeatability of the drifter deployments (Table X in the supplement). Our calculations of the Pearson correlation coefficients confirm, that the two pressure sensors are redundant (Figure 4). Additionally there is a correlation between the pressure sensors and the Magnetometer Y readings. The other sensor modalities represent independent variables.

**3.3**

~~Distance and similarity measures were used to test the repeatability of the datasets. The probability density distribution of each time series data set was compared with the ensemble probability density distributions of all data sets. Various measures can then give an indication on how much the two compared probability density distributions equal each other, meaning how similar they are. Zero values indicate that the distributions are similar and that the experiment is repeatable. For this study, the Chi Squared Error, the Kullback Leibler divergence (KLD), mean average error, the mean squared error and the data range normalized root mean square were calculated for every sensor modality and direction (n = 40). The results are provided in Table ??. The values are generally very low, indicating that the empirical probability density distributions of the single deployments do not deviate much from the probability density distributions of the whole dataset. The highest KLD values are the ones of the right pressure, the magnetometer in x- and z-direction, the acceleration in y-direction and the quaternions in w-direction. The values are however still very close to zero, thus indicating a high repeatability of the drifter deployments.~~

~~Ensemble statistical measures of comparison for all sensor modes. Shown are Chi Squared error (Chi), Kullback Leibler divergence (KLD), mean average error (MAE), mean squared error (MSD) and data range normalized root mean square (RMSD). Sensor mode Mean ChiMean KLD Mean MAE Mean MSD Mean RMSDPressure left Pressure right Magnetometer X Magnetometer Y Magnetometer Z Accelerometer X Accelerometer Y Accelerometer Z Gyroscope X Gyroscope Y Gyroscope Z Quaternion X Quaternion Y Quaternion Z Quaternion W~~

**3.3**

**Table 2.** Estimated multi-modal sample sizes for ±10% precision and a 95% confidence interval based on measured mean values and standard deviations from all deployments (n=40), as well as estimated sample sizes for supraglacial and subglacial deployments based on the utility rate and the measured mean values and standard deviations.

| Sensor mode |  Required sample size estimate | Supraglacial | Subglacial |
|---|---|---|---|
| Pressure left | 2 | 3 | 4 |
| Pressure right | 2 | 3 | 4 |
| Magnetometer X | 1264 | 1732 | 2180 |
| Magnetometer Y | 531 | 728 | 916 |
| Magnetometer Z | 296 | 406 | 511 |
| ‖Magnetometer‖ | 3 | 4 | 5 |
| Accelerometer X | 479,603 | 656,991 | 826,902 |
| Accelerometer Y | 7259 | 9944 | 12,516 |
| Accelerometer Z | 1,382,976 | 1,894,488 | 2,384,442 |
| ‖Accelerometer‖ | 670 | 918 | 1155 |
| Gyroscope X | 115,419 | 158,109 | 198,999 |
| Gyroscope Y | 14,309,576 | 19,602,159 | 24,671,683 |
| Gyroscope Z | 301,182 | 412,578 | 519,280 |
| ‖Gyroscope‖ | 281 | 385 | 485 |

5 ~~Calculating the Pearson correlation coefficients between the different sensor datasets establishes potential correlations between the different sensor modalities and directions and can thus indicate if modalities are redundant or if they represent independent variables. To classify the associations, a modified classification scheme from Cohen (1992) was used. Thereby correlation coefficients from -1.0 to -0.9 and 0.9 to 1.0 were classified as very strong association. Coefficients from -0.9 to -0.5 and 0.5 to 0.9 as strong association, coefficients from -0.5 to -0.3 and 0.3 to 0.5 as moderate association, from -0.3 to -0.07 and 0.07 to 0.3 as weak association and correlation coefficients between -0.07 and 0.07 were classified as not associated. The resulting correlation coefficients together with their association classifications are shown in Figure 4. This figure confirms that the two pressure sensors are indeed redundant. The results also show an interesting correlation between the lateral pressure sensors and the magnetometer in the y-direction. A moderate negative association exists between the magnetometer in the y- and~~

5

**3.3 Moving mean analysis and velocities**

[revised manuscript text omitted]

Fountain, A. G.: Geometry and Flow Conditions of Subglacial Water at South Cascade Glacier, Washington State, U.S.A.; an Analysis of

35  Tracer Injections, Journal of Glaciology, 39, 143–156, https://doi.org/10.3189/S0022143000015793, 1993.

Germain, S. L. S. and Moorman, B. J.: The Development of a Pulsating Supraglacial Stream, Annals of Glaciology, 57, 31–38, https://doi.org/10.1017/aog.2016.16, 2016.

Gleason, C. J., Smith, L. C., Chu, V. W., Legleiter, C. J., Pitcher, L. H., Overstreet, B. T., Rennermalm, A. K., Forster, R. R., and Yang, K.: Characterizing Supraglacial Meltwater Channel Hydraulics on the Greenland Ice Sheet from in Situ Observations, Earth Surface Processes and Landforms, 41, 2111–2122, https://doi.org/10.1002/esp.3977, 2016.

Goodman, D. J., King, G. C. P., Millar, D. H. M., and Robin, G. d. Q.: Pressure-Melting Effects in Basal Ice of Tem-

5   perate Glaciers: Laboratory Studies and Field Observations Under Glacier D'Argentière, Journal of Glaciology, 23, 259–271, https://doi.org/10.1017/S0022143000029889, 1979.

Gulley, J., Benn, D., Müller, D., and Luckman, A.: A Cut-and-Closure Origin for Englacial Conduits in Uncrevassed Regions of Polythermal Glaciers, Journal of Glaciology, 55, 66–80, https://doi.org/10.3189/002214309788608930, 2009.

Haldorsen, S.: Grain-Size Distribution of Subglacial till and Its Realtion to Glacial Scrushing and Abrasion, Boreas, 10, 91–105,

10  https://doi.org/10.1111/j.1502-3885.1981.tb00472.x, 1981.

Hantz, D. and Llioutry, L.: Waterways, Ice Permeability at Depth, and Water Pressures at Glacier D'Argentière, French Alps, Journal of Glaciology, 29, 227–239, https://doi.org/10.3189/S0022143000008285, 1983.

Harper, J. T., Bradford, J. H., Humphrey, N. F., and Meierbachtol, T. W.: Vertical Extension of the Subglacial Drainage System into Basal Crevasses, Nature, 467, 579, 2010.

15  Hart, J. K., Martinez, K., Ong, R., Riddoch, A., Rose, K. C., and Padhy, P.: A Wireless Multi-Sensor Subglacial Probe: Design and Preliminary Results, Journal of Glaciology, 52, 389–397, https://doi.org/10.3189/172756506781828575, 2006.

Hart, J. K., Rose, K. C., and Martinez, K.: Subglacial till Behaviour Derived from in Situ Wireless Multi-Sensor Sub-glacial Probes: Rheology, Hydro-Mechanical Interactions and till Formation, Quaternary Science Reviews, 30, 234–247, https://doi.org/10.1016/j.quascirev.2010.11.001, 2011a.

20 Hart, J. K., Rose, K. C., Waller, R. I., Vaughan-Hirsch, D., and Martinez, K.: Assessing the Catastrophic Break-up of Briksdalsbreen, Norway, Associated with Rapid Climate Change, Journal of the Geological Society, 168, 673–688, https://doi.org/10.1144/0016-76492010-024, 2011b.

Hart, J. K., Rose, K. C., Clayton, A., and Martinez, K.: Englacial and Subglacial Water Flow at Skálafellsjökull, Iceland Derived from Ground Penetrating Radar, in Situ Glacsweb Probe and Borehole Water Level Measurements, Earth Surface Processes and Landforms,
25 40, 2071–2083, https://doi.org/10.1002/esp.3783, 2015.

Hart, J. K., Martinez, K., Basford, P. J., Clayton, A. I., Robson, B. A., and Young, D. S.: Surface Melt Driven Summer Diurnal and Winter Multi-Day Stick-Slip Motion and till Sedimentology, Nature Communications, 10, 1599, https://doi.org/10.1038/s41467-019-09547-6, 2019.

Hasnain, S. I., Jose, P. G., Ahmad, S., and Negi, D. C.: Character of the Subglacial Drainage System in the Ablation Area of Dokriani Glacier,
30 India, as Revealed by Dye-Tracer Studies, Journal of Hydrology, 248, 216–223, https://doi.org/10.1016/S0022-1694(01)00404-8, 2001.

Hou, H., Deng, Z., Martinez, J., Fu, T., Duncan, J., Johnson, G., Lu, J., Skalski, J., Townsend, R., and Tan, L.: A Hydropower Biological Evaluation Toolset (HBET) for Characterizing Hydraulic Conditions and Impacts of Hydro-Structures on Fish, Energies, 11, 990, 2018.

How, P., Benn, D. I., Hulton, N. R. J., Hubbard, B., Luckman, A., Sevestre, H., van Pelt, W. J. J., Lindbäck, K., Kohler, J., and Boot, W.: Rapidly Changing Subglacial Hydrological Pathways at a Tidewater Glacier Revealed through Simultaneous Observations of Water
35 Pressure, Supraglacial Lakes, Meltwater Plumes and Surface Velocities, The Cryosphere, 11, 2691–2710, https://doi.org/10.5194/tc-11-2691-2017, 2017.

Hubbard, B. and Nienow, P.: Alpine Subglacial Hydrology, Quaternary Science Reviews, 16, 939–955, 1997.

Hubbard, B. P., Sharp, M. J., Willis, I. C., Nielsen, M. K., and Smart, C. C.: Borehole Water-Level Variations and the Structure of the Subglacial Hydrological System of Haut Glacier d'Arolla, Valais, Switzerland, Journal of Glaciology, 41, 572–583, https://doi.org/10.3189/S0022143000034894, 1995.

Iken, A.: Measurements of Water Pressure in Moulins as Part of a Movement Study of the White Glacier, Axel Heiberg Island, Northwest
5 Territories, Canada, Journal of Glaciology, 11, 53–58, https://doi.org/10.3189/S0022143000022486, 1972.

Iken, A. and Bindschadler, R. A.: Combined Measurements of Subglacial Water Pressure and Surface Velocity of Finde-lengletscher, Switzerland: Conclusions about Drainage System and Sliding Mechanism, Journal of Glaciology, 32, 101–119, https://doi.org/10.3189/S0022143000006936, 1986.

Isenko, E., Naruse, R., and Mavlyudov, B.: Water Temperature in Englacial and Supraglacial Channels: Change along
10 the Flow and Contribution to Ice Melting on the Channel Wall, Cold Regions Science and Technology, 42, 53–62, https://doi.org/10.1016/j.coldregions.2004.12.003, 2005.

Iverson, N. R., Hooyer, T. S., Fischer, U. H., Cohen, D., Moore, P. L., Jackson, M., Lappegard, G., and Kohler, J.: Soft-Bed Experiments beneath Engabreen, Norway: Regelation Infiltration, Basal Slip and Bed Deformation, Journal of Glaciology, 53, 323–340, https://doi.org/10.3189/002214307783258431, 2007.

15 Jaffe, J. S., Franks, P. J. S., Roberts, P. L. D., Mirza, D., Schurgers, C., Kastner, R., and Boch, A.: A Swarm of Autonomous Miniature Underwater Robot Drifters for Exploring Submesoscale Ocean Dynamics, Nature Communications, 8, 14 189, https://doi.org/10.1038/ncomms14189, 2017.

Jarosch, A. H. and Gudmundsson, M. T.: A Numerical Model for Meltwater Channel Evolution in Glaciers, The Cryosphere, 6, 493–503, https://doi.org/10.5194/tc-6-493-2012, 2012.

20 Kor, P. S. G., Shaw, J., and Sharpe, D. R.: Erosion of Bedrock by Subglacial Meltwater, Georgian Bay, Ontario: A Regional View, Canadian Journal of Earth Sciences, 28, 623–642, https://doi.org/10.1139/e91-054, 1991.

Kriewitz-Byun, C. R., Tuthan, J. A., Gert, T., Albayrak, I., Kammerer, S., Vetsch, D. F., Peter, A., Stoltz, U., Gabl, W., and Marbacher, D.: Research Overview on Multi-Species Downstream Migration Measures at the Fithydro Test Case HPP Bannwil, in: 12th International Symposium on Ecohydraulics (ISE 2018), 2018.

25 Kullback, S. and Leibler, R. A.: On Information and Sufficiency, The Annals of Mathematical Statistics, 22, 79–86, https://doi.org/10.1214/aoms/1177729694, 1951.

Landon, K. C., Wilson, G. W., Özkan-Haller, H. T., and MacMahan, J. H.: Bathymetry Estimation Using Drifter-Based Velocity Measurements on the Kootenai River, Idaho, Journal of Atmospheric and Oceanic Technology, 31, 503–514, https://doi.org/10.1175/JTECH-D-13-00123.1, 2014.

30 Lin, Z., Xiong, Y., Dai, H., and Xia, X.: An Experimental Performance Evaluation of the Orientation Accuracy of Four Nine-Axis MEMS Motion Sensors, in: 2017 5th International Conference on Enterprise Systems (ES), pp. 185–189, https://doi.org/10.1109/ES.2017.37, 2017.

Lishman, B., Wadham, J., Drinkwater, B., Kendall, J.-M., Burrow, S., Hilton, G., and Craddock, I.: Assessing the Utility of Acoustic Communication for Wireless Sensors Deployed beneath Ice Sheets, Annals of Glaciology, 54, 124–134, https://doi.org/10.3189/2013AoG64A022, 35 2013.

Lock, G. S. H.: The Growth and Decay of Ice, Cambridge University Press, 1990.

Marchant, R., Reading, D., Ridd, J., Campbell, S., and Ridd, P.: A Drifter for Measuring Water Turbidity in Rivers and Coastal Oceans, Marine Pollution Bulletin, 91, 102–106, https://doi.org/10.1016/j.marpolbul.2014.12.021, 2015.

Martinez, K., Hart, J. K., and Ong, R.: Environmental Sensor Networks, Computer, 37, 50–56, https://doi.org/10.1109/MC.2004.91, 2004.

Martinez, K., Hart, J. K., Basford, P. J., Bragg, G. M., Ward, T., and Young, D. S.: A Geophone Wireless Sensor Network for Investigating Glacier Stick-Slip Motion, Computers & Geosciences, 105, 103–112, https://doi.org/10.1016/j.cageo.2017.05.005, 2017.

Meire, L., Mortensen, J., Meire, P., Juul-Pedersen, T., Sejr, M. K., Rysgaard, S., Nygaard, R., Huybrechts, P., and Meysman, 5 F. J. R.: Marine-Terminating Glaciers Sustain High Productivity in Greenland Fjords, Global Change Biology, 23, 5344–5357, https://doi.org/10.1111/gcb.13801, 2017.

Nienow, P., Sharp, M., and Willis, I.: Seasonal Changes in the Morphology of the Subglacial Drainage System, Haut Glacier d'Arolla, Switzerland, Earth Surface Processes and Landforms, 23, 825–843, https://doi.org/10.1002/(SICI)1096-9837(199809)23:9<825::AID-ESP893>3.0.CO;2-2, 1998.

10 Oroza, C., Tinka, A., Wright, P. K., and Bayen, A. M.: Design of a Network of Robotic Lagrangian Sensors for Shallow Water Environments with Case Studies for Multiple Applications, Proceedings of the Institution of Mechanical Engineers, Part C: Journal of Mechanical Engineering Science, 227, 2531–2548, https://doi.org/10.1177/0954406213475947, 2013.

Postacchini, M., Centurioni, L. R., Braasch, L., Brocchini, M., and Vicinanza, D.: Lagrangian Observations of Waves and Currents From the River Drifter, IEEE Journal of Oceanic Engineering, 41, 94–104, https://doi.org/10.1109/JOE.2015.2418171, 2016.

15 Rada, C. and Schoof, C.: Channelized, Distributed, and Disconnected: Subglacial Drainage under a Valley Glacier in the Yukon, The Cryosphere, 12, 2609–2636, https://doi.org/10.5194/tc-12-2609-2018, 2018.

Reverdin, G., Boutin, J., Martin, N., Lourenco, A., Bouruet-Aubertot, P., Lavin, A., Mader, J., Blouch, P., Rolland, J., Gaillard, F., and Lazure, P.: Temperature Measurements from Surface Drifters, Journal of Atmospheric and Oceanic Technology, 27, 1403–1409, https://doi.org/10.1175/2010JTECHO741.1, 2010.

20    Rose, K. C., Hart, J. K., and Martinez, K.: Seasonal Changes in Basal Conditions at Briksdalsbreen, Norway: The Winter–Spring Transition, Boreas, 38, 579–590, https://doi.org/10.1111/j.1502-3885.2008.00079.x, 2009.

Röthlisberger, H.: Water Pressure in Intra- and Subglacial Channels, Journal of Glaciology, 11, 177–203, https://doi.org/10.3189/S0022143000022188, 1972.

Schoof, C.: Ice-Sheet Acceleration Driven by Melt Supply Variability, Nature, 468, 803, https://doi.org/10.1038/nature09618, 2010.

25    Schuler, T. V. and Fischer, U. H.: Modeling the Diurnal Variation of Tracer Transit Velocity through a Subglacial Channel, Journal of Geophysical Research, 114, https://doi.org/10.1029/2008JF001238, 2009.

Seaberg, S. Z., Seaberg, J. Z., Hooke, R. L., and Wiberg, D. W.: Character of the Englacial and Subglacial Drainage System in the Lower Part of the Ablation Area of Storglaciären, Sweden, as Revealed by Dye-Trace Studies, Journal of Glaciology, 34, 217–227, https://doi.org/10.3189/S0022143000032263, 1988.

30    Smeets, C. J. P. P., Boot, W., Hubbard, A., Pettersson, R., Wilhelms, F., Broeke, M. R. V. D., and Wal, R. S. W. V. D.: A Wireless Subglacial Probe for Deep Ice Applications, Journal of Glaciology, 58, 841–848, https://doi.org/10.3189/2012JoG11J130, 2012.

Stearns, L. A. and van der Veen, C. J.: Friction at the Bed Does Not Control Fast Glacier Flow, Science, 361, 273–277, https://doi.org/10.1126/science.aat2217, 2018.

Stockdale, R. J., McLelland, S. J., Middleton, R., and Coulthard, T. J.: Measuring River Velocities Using GPS River Flow Tracers (GRiFTers),
35    Earth Surface Processes and Landforms, 33, 1315–1322, https://doi.org/10.1002/esp.1614, 2008.

Stone, D. B. and Clarke, G. K. C.: In Situ Measurements of Basal Water Quality and Pressure as an Indicator of the Character of the Subglacial Drainage Systems, Hydrological Processes, 10, 615–628, https://doi.org/10.1002/(SICI)1099-1085(199604)10:4<615::AID-HYP395>3.0.CO;2-M, 1996.

Sundal, A. V., Shepherd, A., Nienow, P., Hanna, E., Palmer, S., and Huybrechts, P.: Melt-Induced Speed-up of Greenland Ice Sheet Offset by Efficient Subglacial Drainage, Nature, 469, 521, 2011.

Swift, D. A., Nienow, P. W., and Hoey, T. B.: Basal Sediment Evacuation by Subglacial Meltwater: Suspended Sediment Transport from Haut Glacier d'Arolla, Switzerland, Earth Surface Processes and Landforms, 30, 867–883, https://doi.org/10.1002/esp.1197, 2005.

5    TE connectivity sensors: MS5837-02BA: Ultra-Small Gel Filled Pressure & Temperature Sensor, with Stainless Steel Cap, https://www.te.com/global-en/product-CAT-BLPS0059.html, 2017.

Tinka, A., Strub, I., Wu, Q., and Bayen, A. M.: Quadratic Programming Based Data Assimilation with Passive Drifting Sensors for Shallow Water Flows, in: Proceedings of the 48h IEEE Conference on Decision and Control (CDC) Held Jointly with 2009 28th Chinese Control Conference, pp. 7614–7620, https://doi.org/10.1109/CDC.2009.5399663, 2009.

10    Tinka, A., Rafiee, M., and Bayen, A. M.: Floating Sensor Networks for River Studies, IEEE Systems Journal, 7, 36–49, https://doi.org/10.1109/JSYST.2012.2204914, 2013.

Urbanski, J. A., Stempniewicz, L., Węsławski, J. M., Dragańska-Deja, K., Wochna, A., Goc, M., and Iliszko, L.: Subglacial Discharges Create Fluctuating Foraging *Hotspots* for Sea Birds in Tidewater Glacier Bays, Scientific Reports, 7, 43 999, https://doi.org/10.1038/srep43999, 2017.

15    Van de Wal, R. S. W., Boot, W., Van den Broeke, M. R., Smeets, C., Reijmer, C. H., Donker, J. J. A., and Oerlemans, J.: Large and Rapid Melt-Induced Velocity Changes in the Ablation Zone of the Greenland Ice Sheet, science, 321, 111–113, 2008.

van de Wal, R. S. W., Smeets, C. J. P. P., Boot, W., Stoffelen, M., van Kampen, R., Doyle, S. H., Wilhelms, F., van den Broeke, M. R., Reijmer, C. H., Oerlemans, J., and Hubbard, A.: Self-Regulation of Ice Flow Varies across the Ablation Area in South-West Greenland, The Cryosphere, 9, 603–611, https://doi.org/https://doi.org/10.5194/tc-9-603-2015, 2015.

20    Vatne, G. and Irvine-Fynn, T. D. L.: Morphological Dynamics of an Englacial Channel, Hydrology and Earth System Sciences, 20, 2947–
         2964, https://doi.org/https://doi.org/10.5194/hess-20-2947-2016, 2016.

      Vieli, A., Jania, J., Blatter, H., and Funk, M.: Short-Term Velocity Variations on Hansbreen, a Tidewater Glacier in Spitsbergen, Journal of
         Glaciology, 50, 389–398, https://doi.org/10.3189/172756504781829963, 2004.

      Vivian, R. and Bocquet, G.: Subglacial Cavitation Phenomena Under the Glacier D'Argentière, Mont Blanc, France, Journal of Glaciology,
         12, 439–451, https://doi.org/10.3189/S0022143000031853, 1973.

785   Walder, J. S. and Fowler, A.: Channelized Subglacial Drainage over a Deformable Bed, Journal of Glaciology, 40, 3–15, 1994.

      Weertman, J.: General Theory of Water Flow at the Base of a Glacier or Ice Sheet, Reviews of Geophysics, 10, 287–333,
         https://doi.org/10.1029/RG010i001p00287, 1972.

      Werder, M. A., Hewitt, I. J., Schoof, C. G., and Flowers, G. E.: Modeling Channelized and Distributed Subglacial Drainage in Two Dimen-
         sions, Journal of Geophysical Research: Earth Surface, 118, 2140–2158, https://doi.org/10.1002/jgrf.20146, 2013.

790   Willis, I. C., Sharp, M. J., and Richards, K. S.: Configuration of the Drainage System of Midtdalsbreen, Norway, as Indicated by Dye-Tracing
         Experiments, Journal of Glaciology, 36, 89–101, https://doi.org/10.3189/S0022143000005608, 1990.

      Willis, I. C., Fitzsimmons, C. D., Melvold, K., Andreassen, L. M., and Giesen, R. H.: Structure, Morphology and Water Flux of a Subglacial
         Drainage System, Midtdalsbreen, Norway, Hydrological Processes, 26, 3810–3829, https://doi.org/10.1002/hyp.8431, 2012.

      Xanthidis, M., Li, A. Q., and Rekleitis, I.: Shallow Coral Reef Surveying by Inexpensive Drifters, in: OCEANS 2016 - Shanghai, pp. 1–9,
795      https://doi.org/10.1109/OCEANSAP.2016.7485639, 2016.

      Zhao, Y., Görne, L., Yuen, I.-M., Cao, D., Sullman, M., Auger, D., Lv, C., Wang, H., Matthias, R., Skrypchuk, L., and Mouzakitis, A.: An
         Orientation Sensor-Based Head Tracking System for Driver Behaviour Monitoring, Sensors, 17, 2692, https://doi.org/10.3390/s17112692,
         2017.

      Zwally, H. J., Abdalati, W., Herring, T., Larson, K., Saba, J., and Steffen, K.: Surface Melt-Induced Acceleration of Greenland Ice-Sheet
800      Flow, Science, 297, 218–222, 2002.

---

## Author Response (AR2)

**Response to referee and editor comments**

**Pressure and inertia sensing drifters for glacial hydrology flow path measurements**

Andreas Alexander[1,2], Maarja Kruusmaa[3,5], Jeffrey A. Tuhtan[3], Andrew J. Hodson[2,4], Thomas V. Schuler[1,2], Andreas Kääb[1]

[1]Department of Geosciences, University of Oslo, 0316 Oslo, Norway
[2]Department of Arctic Geology, The University Centre in Svalbard, 9171 Longyearbyen, Norway
[3]Centre for Biorobotics, Tallinn University of Technology, 12618 Tallinn, Estonia
[4]Department of Environmental Sciences, Western Norway University of Applied Sciences, 6856 Sogndal, Norway
[5]Centre for Autonomous Marine Operations and Systems, Norwegian University of Science and Technology, 7491 Trondheim, Norway

*Correspondence to*: Andreas Alexander (andreas.alexander@geo.uio.no)

We would like to thank the reviewer Samuel Doyle for his second round of additional constructive feedback that helped to further improve this manuscript. We would also like to thank Jan De Rydt for his work editing this manuscript. We have now changed the title of the manuscript and removed the table with the references to wireless sub-surface sensor systems. We further on checked the manuscript for additional spelling and grammatical errors, implemented the suggestions made by the reviewer and included a statement about data availability. The point-by-point response to the reviewer comments are presented below. A mark-up version of the manuscript, showing the changes made in response to the referee's comments follows thereafter.

**Response to Referee 2:**

In the following we respond to the comments made by Samuel Doyle in his second helpful and valuable review and outline how we addressed these in the revision of the manuscript. Referee comments are presented in *italic*, and our replies follow directly thereafter.

*My previous comments have largely been addressed, however, further work is required to justify the inclusion of the table of wireless sub-surface sensor systems (now Table S2; see below). The same applies to Table 1. The manuscript needs a through check for grammar and typographical mistakes. Please see a list of comments below.*

We have now removed Table S2 from both the manuscript and the supplement and removed several typographical and grammar mistakes from the manuscript.

*Title - Consider whether the title could be shortened and made clearer to something like "Multi-sensor drifters for glacial hydrology flow path measurements". I'm not sure many readers will immediately understand precisely what is meant by 'Multi-modal'. Consider what you mean by 'repeatable' in the title, and elsewhere.*

Thanks for pointing this out and you are certainly right with your observations regarding the word multi-modal. We have now changed the title of the manuscript to: "Pressure and inertia sensing drifters for glacial hydrology flow path measurements".

*P1L6 - you need to make clear that the experiments were repeated multiple times in the same channel section and then the results compared.*

We specified it accordingly.

*P1L16 - '... repeated measurements ...'*

Thanks, we changed this.

*Table 1 - Please consider whether this table is essential to the reader. I'm not sure that it is necessary here and It could be moved to the supplement. The reader does not currently have access to an example dataset so does not need the metadata. Even if an example dataset were provided the metadata does not need to be provided in the manuscript. Column names and abbreviations should be explained.*

Yes, you are indeed right. The table does not add additional value to this manuscript and we therefore removed it.

*P2L17 - the reference here to Table S2 which lists subsurface sensors is a bit misleading as it is immediately followed by a list of other techniques without explaining the difference. Table S2 also focusses on wireless sub-surface glacial sensors*

*neglecting entirely (without a reason being given) very similar wired sensor systems, which are used more commonly. Neither appear to be directly relevant to this study. More work is required to organise this section for the readers benefit.*

We agree and removed Table S2 from the Supplementary material, as well as its' reference from the text.

*P7L15 - what kind of debris? I'm guessing you mean ice?*

No, we meant rocks, as in for example "debris covered glaciers". We changed it to "rock debris" in the manuscript for clarification.

*P7L16 - Suggest 'emerging' or 're-emerging'.*

Thanks for this suggestion, we changed it accordingly.

*P11L5 - the equations for the recovery rate etc are methods and should be moved to the methods section. It's not necessary to show your workings. These results could potentially be presented in a table.*

We moved the equations to the Methods section and presented the results in a table. We also removed the workings.

*P11L30 - here a list is started using a colon but then the second part of the list follows as a new sentence. Suggest avoid new sentence.*

We avoided the new sentence.

*P12L8 - Table S3 not A3.*

Thanks for catching this mistake. We changed it and it is now Table S2, as we removed the original Table S1. See response to the first comment.

*P13L3 - increasing atmospheric pressure with (known) decrease in altitude could be calculated to isolate its contribution here.*

We agree, however the sensors deployed in this work were designed to be submerged. In this case the changes in depth / dynamic water pressure are much larger than the local change in atmospheric pressure. But it is planned that future drifters include a barometric pressure sensor as well to estimate elevation changes over larger distances.

*P13L9 - ensure consistency with writing out confidence interval in full and abbreviating it*

Thanks for pointing this out. We now only use the abbreviation after we introduced it in the text.

*P14L1 - the sentence beginning (on this page) '... all sensor modalities' needs revising.*

We revised it.

*P14L10 - its the hydrodynamics of the drifter and balloon than lead the drifter to face away, combined with the current.*

We adjusted this sentence accordingly.

*P19L11 - is the use of the double greater than sign >> intended to indicate P approximately greater than 0.05?*

It means much greater than 0.05.

*P19L14 - 'could' instead of 'can'*

Thanks, we changed it.

*P20L29 - I'm not sure this sentence says what you intend it to. Do you mean that additional fieldwork should compare drifter data to other measurements (e.g. tracer velocity)? Please clarify.*

Thanks for the comment. We clarified this sentence.

*P21L8 - time-averaged?*

Yes, you are indeed correct. Thanks for catching this mistake.

*P21L10 – typo*

We corrected it and changed the word order of the sentence.

5   *P21L11 - sentence beginning 'After low pass filtering ...' needs re-ordering.*

Thanks. We re-ordered it.

*Ensure Table S2 and Figure S1 are referenced in the manuscript. If not, are they necessary?*

We referenced Table S2 in the text (now referenced as Table S1) and removed Figure S1 from the supplementary material.

[revised manuscript text omitted]

---

## Author Response (AR3)

**Response to editor comments**

**Pressure and inertia sensing drifters for glacial hydrology flow path measurements**

Andreas Alexander[1,2], Maarja Kruusmaa[3,5], Jeffrey A. Tuhtan[3], Andrew J. Hodson[2,4], Thomas V. Schuler[1,2], Andreas Kääb[1]

[1]Department of Geosciences, University of Oslo, 0316 Oslo, Norway
[2]Department of Arctic Geology, The University Centre in Svalbard, 9171 Longyearbyen, Norway
[3]Centre for Biorobotics, Tallinn University of Technology, 12618 Tallinn, Estonia
[4]Department of Environmental Sciences, Western Norway University of Applied Sciences, 6856 Sogndal, Norway
[5]Centre for Autonomous Marine Operations and Systems, Norwegian University of Science and Technology, 7491 Trondheim, Norway

*Correspondence to*: Andreas Alexander (andreas.alexander@geo.uio.no)

We would like to thank the editor Jan De Rydt for his comments to our latest revision of the manuscript. We have addressed the comments and our response to the editor comments is presented below. A mark-up version of the manuscript, showing the changes made in response to the editor's comments follows thereafter.

**Response to editor comments:**

In the following we respond to the comments made by Jan De Rydt and outline how we addressed these in the revision of the manuscript. Editor comments are presented in *italic*, and our replies follow directly thereafter.

*Thank you for providing a revised manuscript and for your comprehensive treatment of the reviewer's suggestions. I have some minor comments that you might want to consider as part of your final manuscript submission.*

Thank you for providing these valuable comments. We have addressed and implemented them all.

*1. Recovery and utility rates are now defined in the Methods section, but this definition would benefit from a short motivation. At the moment these are just 3 floating equations without context.*

We have addressed this suggestion by adding a short motivation (P9 L11-15).

*2. Please expand on why the data is not publicly available in the 'Data Availability' section. Please note the guidelines in the Copernicus data policy booklet in this context: "If the data are not publicly accessible, a detailed explanation of why this is the case is required (e.g. applicable laws, university and research institution policies, funder terms, privacy, intellectual property and licensing agreements, and the ethical context of the research);"*

Thanks for reminding us of the guidelines. We have now made all raw data publicly available and added the doi to the data in the manuscript (P21 L18). The data can be found under doi: 10.5281/zenodo.3660488

[revised manuscript text omitted]